# Generating Boundary Conditions for Compound Flood Modeling in a Probabilistic Framework

Pravin Maduwantha[1,2], Thomas Wahl[1,2], Sara Santamaria-Aguilar[1,2], Robert Jane[1,2], Sönke Dangendorf[3], Hanbeen Kim[4,5], and Gabriele Villarini[4,5]

[1] Department of Civil, Environmental and Construction Engineering, University of Central Florida, Orlando, FL 32816, USA
[2] National Center for Integrated Coastal Research, University of Central Florida, Orlando, FL 32816, USA
[3] Department of River-Coastal Science and Engineering, Tulane University, 6823 St. Charles Avenue, New Orleans, LA, 70118-5698, USA
[4] Department of Civil and Environmental Engineering, Princeton University, Princeton, NJ 08544, USA
[5] High Meadows Environmental Institute, Princeton University, Princeton, NJ 08544, USA

*Correspondence to*: Pravin Maduwantha (pravin@ucf.edu)

**Abstract.** Compound flood risk assessments require probabilistic estimates of flood depths and extents that are derived from compound flood models. It is essential to simulate a wide range of flood driver conditions to capture the full range of variability in resultant flooding. Although recent advancements in computational resources and the development of faster compound flood models allow for more rapid simulations, generating a large enough set of storm events for boundary conditions remains a challenge. In this study, we introduce a statistical framework designed to generate many synthetic but physically plausible compound events, including storm-tide hydrographs and rainfall fields, which can serve as boundary conditions for dynamic compound flood models. We apply the proposed framework to Gloucester City in New Jersey, as a case study. The results demonstrate its effectiveness in producing synthetic events covering the unobserved regions of the parameter space. We use flood model simulations to assess the importance of explicitly accounting for variability in mean sea level (MSL) and tides in generating the boundary conditions. Results highlight that MSL anomalies and tidal conditions alone can lead to differences in flood depths exceeding 1 m and 1.2 m, respectively, in parts of Gloucester City. While we use historically observed events, the framework can be applied to model output data including hindcasts or future projections.

## 1 Introduction

Flooding in coastal regions can be caused by various hydrometeorological drivers such as precipitation, excess river discharge, wind-driven storm surge, mean sea level (MSL), and high tides. When these flood drivers occur simultaneously or in close succession, they often lead to compound flooding, which can result in more severe flood impacts and substantial socioeconomic losses (e.g., Hendry et al., 2019; Nasr et al., 2023; Wahl et al., 2015; Ward et al., 2018). Therefore, accurately quantifying and characterizing compound flood risk is crucial for effective flood risk management and mitigation, infrastructure design, urban planning, the (re)insurance markets, emergency response, and more.

Flood depths (and extents) are typically estimated using compound flood models, due to the scarcity of data on historic flood events, e.g., from high water marks or satellite observations. One simple approach is to use static compound flood models (also referred to as 'bathtub' models) (e.g., Gallien, 2016; Seenath et al., 2016; Semmendinger et al., 2021), but these models tend to overestimate flood extent primarily due to the assumption that peak water levels are maintained indefinitely and by neglecting critical factors such as bottom friction and flood duration (Barnard et al., 2019; Breilh et al., 2013; Gallien, 2016; Kumbier et al., 2019). Alternatively, dynamic compound flood models are employed to capture the physical mechanisms of coastal and inland flooding, and they have been shown to provide good results for various terrain types, catchment sizes, and flood driver combinations (Kumbier et al., 2019; Lewis et al., 2013; Ramirez et al., 2016; Vousdoukas et al., 2016). However, dynamic compound flood models require time series of the different flood drivers, and their relative timing to each other, as boundary conditions.

Temporally and spatially varying boundary conditions permit a thorough exploration of different scenarios, including variations in timing, intensity, and spatial extent of flood drivers (Harrison et al., 2022; Quinn et al., 2014). The development of faster compound flood models (e.g. SFINCS (Super-Fast INundation of CoastS)) coupled with the increase in computational resources enables many scenarios to be rapidly propagated through dynamic compound flood models. The scarcity of long-term concurrent observational records of flood drivers poses a challenge in generating plausible extreme conditions that can serve as boundary conditions for those models (Ward et al., 2018). One way of addressing this issue is by using physics-based models to generate many events (e.g. rainfall-surge-discharge events) (Bass and Bedient, 2018; Bates et al., 2021; Gori et al., 2020; Nederhoff et al., 2024; Orton et al., 2020). For example, Gori et al. (2020) first derived synthetic tropical cyclone (TC) tracks and then simulated the resultant rainfall (RF) fields using a physics-based model and the associated storm tides through a hydrodynamic model (ADvanced CIRCulation (ADCIRC) model (Luettich R. A., 1992)). These RF fields and storm tides were subsequently used as boundary conditions in a one-way coupled hydrodynamic modeling framework to simulate the total flood levels in a tidal estuary.

Generating boundary conditions via physics-based modeling is often computationally expensive, thus making it challenging to implement across diverse climate and environmental conditions. Statistical approaches offer a computationally cheaper alternative by modeling the joint probability distribution of flood drivers directly and simulating scenarios from the fitted distribution model. These scenarios are then propagated through compound flood models, allowing for the assessment of flood impacts while reducing computational demands compared to more complex physical models. Bayesian networks (e.g., Couasnon et al., 2018), bivariate logistic models (e.g., Serafin et al., 2019), and copulas (e.g., Liu et al., 2024; Moftakhari et al., 2019; Zellou and Rahali, 2019) are examples of statistical approaches applied to analyze compound flood drivers. These approaches still possess various limitations when deriving time series of boundary conditions. For instance, they often rely on a representative event (e.g., Liu et al., 2024) or a simplistic sinusoidal shape (e.g., Moftakhari et al., 2019) of the hydrographs for all simulations, which oversimplifies the temporal variability of flood drivers. They also may neglect the timing dynamics between RF-runoff and storm tides, either assuming both flood drivers peak simultaneously or assuming a range of possible

time lags (e.g., Moftakhari et al., 2019). Furthermore, they fail to capture the spatial variability of RF fields as they rely on RF point data from observations or models (e.g., Zellou & Rahali, 2019).

Harrison et al. (2022) highlighted that in both large and small estuaries, storm surge intensity rather than height was the main flooding driver, while Shen et al. (2019) noted that longer-duration storm tides led to greater backward flow volumes in underground pipes. Therefore, generating realistic synthetic storm-tide hydrographs is crucial since the flood extent, particularly around the peak water level, is often highly sensitive to the shape of the storm-tide hydrograph (Quinn et al., 2014). Methods for generating extreme storm-tide hydrographs can be mainly categorized into deterministic and stochastic. Deterministic methods use pre-defined shapes or observed event patterns, such as triangles (e.g., Vousdoukas et al., 2016) or sinusoidal functions (e.g., Moftakhari et al., 2019), which simplify the event structure but may not capture the natural asymmetry in water level profiles. This approach, although efficient, may also ignore nonlinear interactions between tides and surges, depending on how the method is applied (Arns et al., 2020). Rescaling of total water level (or non-tidal residuals (NTR) time series) of observed events is another deterministic approach that leverages observed event data (Dawson et al., 2005; Kim et al., 2023; Xu et al., 2024). This method incorporates site-specific information, eliminating the assumption of symmetrical rising and falling limbs in the total water level profile. Alternatively, stochastic simulation methods can be used to generate many physically plausible events for a given peak water level, while accounting for natural temporal variability in storm tides (MacPherson et al., 2019; Wahl et al., 2011, 2012). For example, Wahl et al. (2011) parameterized water levels around peak tides using 19 sea level and six time parameters, fitting independent marginal distributions to each, and modeling dependencies through linear regression. Filters were applied to ensure realistic event generation, effectively recreating the peak water level–intensity relationship observed at German Bight tide gauges. Dullaart et al. (2023) created a global dataset of storm tide hydrographs from the depth-averaged hydrodynamic Global Tide and Surge Model (GTSM) (Hersbach et al., 2020). They incorporated nonlinear tide-surge interactions in the surge series by calculating it as the difference in elevation between storm tide simulations and tide-only simulations. However, they assumed that the surge maximum coincided with the high tide.

A variety of methods are available for generating design hyetographs for point RF estimates, ranging from simple geometric shapes (e.g., Chow et al., 1988) to more sophisticated multi-site stochastic models (e.g., Evin et al., 2018). However, relatively few studies have focused on generating synthetic space-time varying RF events. Green et al. (2024) classified methods for simulating space-time varying RF into four main approaches: (1) multi-site temporal simulations (e.g., Brissette et al., 2007; Kleiber et al., 2012), (2) point process theory-based methods (e.g., Burton et al., 2008; Cowpertwait et al., 2002), (3) random field theory-based methods (e.g., Leblois and Creutin, 2013; Papalexiou et al., 2021), and (4) fractal processes in two or three dimensions (e.g., Schertzer and Lovejoy, 1987). These methods are often tailored to specific research objectives depending on their strengths, but they also come with various limitations. For example, while point process theory-based methods are generally robust, they may not accurately capture the complex spatial structures of RF cells (Green et al., 2024). Furthermore, many of these approaches generate stochastic RF fields, without accounting for the temporal dependencies with other flood drivers, such as storm surge, which limits their applicability for generating synthetic compound events.

Among the applications of uniform scaling of flooding drivers, Xu et al. (2024) applied the "same frequency amplification" method to construct a 200-yr storm surge hydrograph and rainfall hyetograph for their flood simulations. However, their approach was limited to point rainfall and assumed uniformly distributed rainfall across the catchment. Kim et al. (2023) proposed a framework for generating synthetic time series of RF fields and associated NTR by scaling time series of observed TC events. The framework was used to capture different spatial patterns of RF fields as this aspect was shown to significantly contribute to compound flood hazard (e.g., Gori et al., 2020). However, their analysis exclusively focused on TC events and the methodology only produces NTR time series and does not extend to producing complete storm-tide hydrographs; this is because it was applied to the Texas coast where the tidal range is small, and where compound flooding is primarily driven by TCs. Other types of storms can produce compound flooding in many other areas and tides often contribute significantly to the resulting still water levels.

The existing statistical approaches that generate time-varying boundary conditions for dynamic compound flood models are primarily intended to construct design events with specified joint return periods (e.g., 50-yr, 100-yr) (Serafin et al., 2019; Moftakhari et al., 2019; Zellou and Rahali, 2019; Kim et al., 2023; Liu et al., 2024; Xu et al., 2024). This method supports the "event-based" flood hazard analysis, where single synthetic events, or a few of them,with known joint return periods are simulated through a flood model, and it is assumed that the joint probability of the flood drivers directly translates into the probability of the flood response. However, this neglects the range of potential different flooding scenarios that may arise from variations in temporal and spatial patterns, differences in the relative timing of multiple flood drivers, and other complex interactions (for example, tide-surge interactions). For a more complete characterization of flood hazard and risk, the flood response of many synthetic events needs to be modeled, allowing the derivation, for example, of return levels of flood depth at all points within the model domain (i.e., "response-based" flood hazard analysis).

In this study, we present a framework for generating many synthetic but physically plausible compound events consisting of storm-tide hydrographs and RF fields that can act as boundary conditions for dynamic compound flood models. We first estimate the joint probability distribution of flood drivers following Maduwantha et al. (2024) and utilize time series of RF and NTR of observed events to generate a synthetic event set. We explicitly account for the intra-annual and longer-term variability of MSL and tides, tide-surge interactions, and relative lag times between the peaks of flood drivers. Then, we use flood model simulations to assess the importance of accounting for MSL and tidal variability in the boundary condition's generation process. We apply the proposed framework for Gloucester City, New Jersey, as a case study.

## 2 Study area

Gloucester City is located in Camden County, New Jersey, and has been impacted by several severe compound flood events in recent years, caused by hurricanes and other intense storms, including Hurricanes Floyd in 1999, Irene in 2011, Sandy in 2012, and an unnamed storm in 2015. The city is bordered by the Delaware River from the west, Newton Creek to the north, and Little Timber Creek to the south exposing the area to flooding from multiple water sources. According to the Federal

Emergency Management Agency (FEMA), a substantial portion of the city's land area falls within designated flood zones, and

over 1,100 residential and commercial properties are exposed to major, severe, or extreme flood risk (Gloucester City New Jersey, 2024; FEMA, 2016). We select the catchment area for Gloucester City comprising two 14-digit hydrologic units (Fig. 1) (Jones et al., 2022).

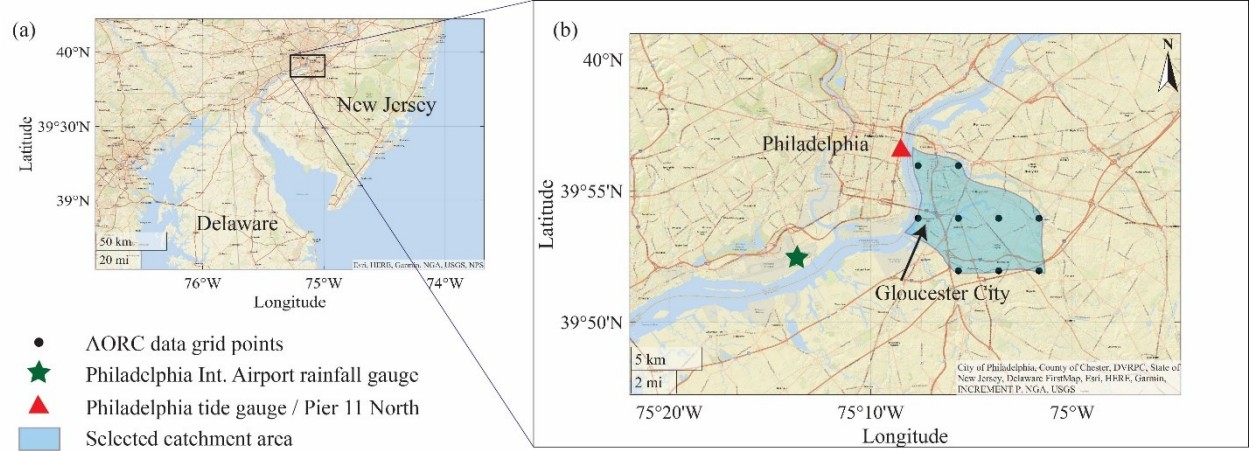

**Figure 1: Location of Gloucester City, selected catchment boundaries, locations of the rainfall gauge, tide gauge, and grid points of the Analysis of Period of Record for Calibration (AORC) data.**

## 3 Data

For the statistical analysis, we consider RF and NTR as flood drivers. We use hourly water level data from the nearest tide gauges to the study site provided by the National Oceanic and Atmospheric Administration (NOAA): Philadelphia (St. ID: 8545240) and Philadelphia Pier 11-north (St. ID: 8545530). The two datasets are merged, adjusting a 1 cm constant offset between the two records during the overlapping period. This results in a 122-year-long dataset from 1901 to 2021 with less than 3% of missing data. The water level time series is then detrended using a 30-day moving average to eliminate the effects

of long-term relative MSL rise and seasonal and interannual MSL variability. Subsequently, a year-by-year harmonic tidal analysis is conducted using the Unified Tidal Analysis and Prediction (UTide) package in MATLAB to determine tidal constituents and tidal levels (Codiga, 2011). Years with more than 25% missing data are removed from the analysis (1903, 1921, 1922, and 1959). We calculate the hourly NTR time series by subtracting the predicted tides from the detrended water levels.

We use both gridded RF data from the Analysis of Period of Record for Calibration (AORC) from 1979 to 2021 and hourly RF gauge data at the Philadelphia International Airport from 1900 to 2021 (Kitzmiller et al., 2018). Although radar-based

quantitative precipitation estimates, such as the Multi-Radar Multi-Sensor (MRMS) products, often provide higher accuracy compared to other gridded rainfall products, their temporal coverage is relatively short (Gao et al., 2021; Gomez et al., 2024). We use AORC rainfall data because of its availability from 1979 and its demonstrated higher accuracy among products with similar temporal coverage (e.g., Hong et al., 2024; Kim and Villarini, 2022), while offering hourly data at ~4 km spatial resolution. Rain gauges measure highly localized rainfall. Assuming that these point measurements occurred uniformly distributed across the entire catchment can misrepresent the compound flood hazard. To address this, we apply a bias correction to the hourly gauge data so it matches the basin-averaged hourly rainfall estimates derived from AORC. This correction is performed using the quantile mapping method, in which both the gauge-based and AORC-based rainfall distributions are fitted to gamma functions (for more details see Maduwantha et al. (2024)).

For identifying TC events, we use the HURDAT2 TC track dataset from the National Hurricane Center, which provides the location of the center of circulation at 6-hour intervals (Landsea and Franklin, 2013). Considering the overlapping periods of available datasets, the joint probability analysis is conducted for the period from 1901 to 2021.

## 4 Methods

The overall methodology to derive synthetic boundary conditions for compound flood inundation modeling with associated annual exceedance probabilities is outlined in the flowchart in Fig. 2. In the following subsections we describe the process in more detail and refer to the relevant boxes (or groups of boxes) in the flowchart for better clarity.

### 4.1 Joint probability estimation

Recent data-driven threshold-selection methods, such as the Sequential Goodness-of-Fit method (Bader et al., 2018), the Extrapolated-Height Stability method (Liang et al., 2019), L-moment ratio stability (Silva Lomba & Fraga Alves, 2020), and a comparative multi-method approach (Radfar et al., 2022), provide robust peak-over-threshold (POT) thresholds but primarily optimize tail fit. Extreme compound flood events are not necessarily generated by extreme flood driver peaks. With favorable timing, duration, and tidal conditions, extreme flooding can occur even under moderate flood-driver conditions (Santamaria et al., 2025). Therefore, we use a two-sided conditional sampling based on the POT approach to identify extreme events, setting NTR and RF thresholds to obtain samples allowing an average of 5 exceedances per year (Jane et al., 2020; Kim et al., 2023). When conditioning on NTR, the maximum RF value within a 3-day window is selected, and the same procedure is followed when conditioning on RF. To ensure independence within the POT samples, a 5-day declustering window (2.5 days before and after the event peaks) is used (Camus et al., 2021). Next, the two conditional samples are stratified into two sets, TC events and non-TC events, using the TC track data set. An event is classified as being caused by a TC if there is a center of circulation within a 350 km radius of the Gloucester City catchment within a 3-day window (2 days before and 1 day after) of a POT event. All other events are categorized as non-TC events. This process is carried out for hourly RF accumulation times from 1 to 48 hours; the RF accumulation time that has the highest correlation with NTR is selected for the subsequent bivariate

statistical analysis. Maduwantha et al. (2024) found significant non-stationarity in Kendall's τ between peak NTR and RF over the analysis period. To capture most recent climate conditions and avoid underestimating compounding effects, we model dependence using only the last 30 years of data. The stratified samples (TC and non-TC) are then fitted to different parametric univariate distributions and copulas to identify the best-fitting marginal distributions and copula families, respectively. Considering the recommendations of Moftakhari et al. (2019) for compound flood assessments, we use the "AND" scenario which represents the exceedance of both variables for calculating annual joint exceedance probabilities (AEPs). The calculated AEPs of two stratified samples are then combined to estimate the final joint probability distribution. To quantify relative joint probabilities along a given isoline, we sample $10^6$ NTR and RF combinations from the fitted copulas, ensuring the proportion of extremes matches the empirical distribution. The relative probability along the isolines is then calculated using a kernel density function, with the "most likely" event assigned to the point of highest relative probability density on the isoline (Salvadori and De Michele, 2013). A more detailed description of the methodology can be found in Maduwantha et al. (2024). We generate an event set of 5,000 combinations of NTR and RF ("target events") by sampling from the fitted copulas such that the relative proportion of extremes is consistent with the empirical distribution. This initial event set contains the peak NTR and peak basin average RF for the selected RF accumulation time reflecting their joint probability of occurrence at the study site (Fig. 2 (e)). In the following subsections, we outline how those peak values are turned into storm-tide hydrographs and temporally varying RF fields with realistic lag times between the peaks of NTR and RF.

## 4.2 Characteristics of NTR and RF time series from TC and non-TC events

Before generating the final synthetic events, we compare the characteristics of TC events and non-TC events to determine whether event generation should be conducted separately for TC and non-TC events or if we can draw time series from the combined dataset, allowing for more variability in the final event set. We extract hourly time series of NTR (Fig. 2 (d)) and hourly RF fields (Fig. 2 (f)) over the Gloucester City catchment during a three-day period around all POT events. This analysis is limited to POT events recorded after January 1979, the start date of the gridded AORC RF data. The joint probability distribution derived in Section 4.1 explicitly accounts for the two different dependence structures between peak NTR and RF for the two different storm types. In this analysis step, we examine the correlations between various characteristics of the time series, including hourly peaks, durations, intensities, and lag times. Additionally, we assess the distributional shapes of peak RF, total RF, RF duration, lag time, NTR duration, and NTR intensity by fitting them to appropriate parametric distributions. We consider Normal, Exponential, Gamma, Lognormal, Birnbaum-Saunders, and generalized Pareto distributions, selecting the best model using the Akaike information criterion (AIC; Akaike, 1974). Previous studies indicate that TCs generally produce more intense RF compared to extratropical cyclones (ETCs), while ETCs often generate longer-duration RF (e.g., Orton et al., 2016; Sinclair et al., 2020). Therefore, we analyze the shapes and durations of the NTR and RF time series from observed events to determine whether there are significant differences between TC and non-TC event time series.

We use a 6-hour continuous dry period to identify independent RF events, and the duration of a given RF event is defined as the non-zero basin average RF to the starting hour of the next 6-hour dry spell. Here we calculate the total RF as the sum of

all the basin-averaged hourly RF quantities of the event. The duration of the NTR events is defined as the duration over which the NTR is continuously above the defined threshold. The intensity of the NTR is calculated as the area under the NTR time series curve above zero within the duration.

### 4.3 The events generation process

 #### 4.3.1 Selecting observed events

To disaggregate the target basin average peak RF spatially and temporally, we select a historical event that closely matches the accumulated RF of the target basin average RF. Given the limited number of observed events, selecting only the nearest event would result in utilizing a single or small number of observed events for all the nearby target scenarios, thereby restricting the diversity of the generated events. Additionally, when the selected RF event is largely different from the target RF, the scaling factor becomes higher and may result in making the synthetic event unrealistic. Therefore, we randomly sample from the observed events, with probabilities defined as the inverse of the difference between target RF and peak basin average RF quantities (of the selected RF accumulation time) of historical events (Fig. 2 (j)). For NTR, we also use the same method for selecting a nearby event using the inverse of the difference between the target NTR and peak hourly NTR of the historical events (Fig. 2 (h)).

 #### 4.3.2 Scaling observed events

We use a similar scaling approach to that introduced by Kim et al. (2023) for assigning time series of data to match target scenarios (peak NTR and peak basin average RF pairs). We calculate the RF scaling factor $K_{RF}$ as follows:

$$K_{RF} = \mathrm{RF}_T / \mathrm{RF}_{obs} \tag{1}$$

where, $\mathrm{RF}_T$ is the target RF and $\mathrm{RF}_{obs}$ is the peak basin average RF (of selected accumulation time) of the selected observed event. Then we multiply the hourly observed RF fields by the scaling factor $K_{RF}$, generating a synthetic RF event with a peak accumulation that matches that of the target RF (Fig. 2 (m)).

For NTR, we calculate the NTR scaling factor $K_{NTR}$ as follows:

$$K_{NTR} = \mathrm{NTR}_T / \mathrm{NTR}_{obs} \tag{2}$$

where, $\mathrm{NTR}_T$ is the target NTR peak and $\mathrm{NTR}_{obs}$ is the peak hourly NTR of the selected observed event. Then we multiply the hourly time series of the NTR by the scaling factor $K_{NTR}$, generating a synthetic NTR event with a peak that matches the peak target NTR (Fig. 2 (k)). Here we only consider the section of the NTR time series for which the NTR is positive around the peak.

#### 4.3.3 Combining scaled NTR time series with tides and MSL

Dynamic compound flood models require total water level time series as boundary conditions which comprise the tide, MSL, and NTR (in some cases also waves, depending on the location). All of those exhibit seasonal variations, which can be

significant and, therefore, cannot be ignored (for NTR this is captured through stratification into TC and non-TC events). As a preliminary step, we assess the variability of MSL and the high and low tides throughout the year, categorized by calendar months. As explained in Maduwantha et al. (2024), we apply a 30-day moving average to the measured water level data to remove any trends before conducting the tidal analysis. We then segregate the 30-day averaged MSL values of the last five years (to ensure that the analysis reflects the most recent conditions) by calendar month. For tides, we extract hourly tidal signal segments spanning 3-day periods around each high tide, covering the last 18.6 years to account for the lunar nodal cycle. These segments are then grouped by calendar month.

To ensure consistency with seasonal variations, we first sample a month based on the distribution of POT observations recorded in each month (i.e., the monthly frequency of occurrence). Target events are derived from copulas fitted to the TC sample and the non-TC sample. If the target event is derived from a copula fitted to TC (non-TC) events, we sample the month from the distribution of TC (non-TC) events (Fig. 2 (a)). Once the month is selected, we randomly sample a MSL value and a tidal signal segment from the selected month (Fig. 2 (g)).

Considering that tide-surge interactions are significant in certain regions, tides, and wind-driven storm surges (here NTR) often show interdependencies. Therefore, it is important to check the variability of the timing of peak NTR relative to tidal levels to determine whether it is necessary to explicitly account for tide-surge interactions when generating synthetic events. Here, we use the observed time difference between peak NTR and the subsequent high tide of the sampled NTR time series to combine it with the sampled tidal signal. Then the sampled MSL value is added, generating the storm tide hydrograph (Fig. 2 (n)).

### 4.3.4 Combining storm tide hydrograph and RF fields

As the final step, the scaled RF fields and calculated storm tide hydrographs are combined to create compound events that can be simulated through a flood model. The timing dynamics of the flood drivers play a vital role in the resultant flood depth (Gori et al. 2020). Therefore, we randomly pick one of the observed lag times (between the peak hourly NTR and peak hourly basin average RF) from the selected NTR event and selected RF event for creating the synthetic compound event (Fig. 2 (p)).

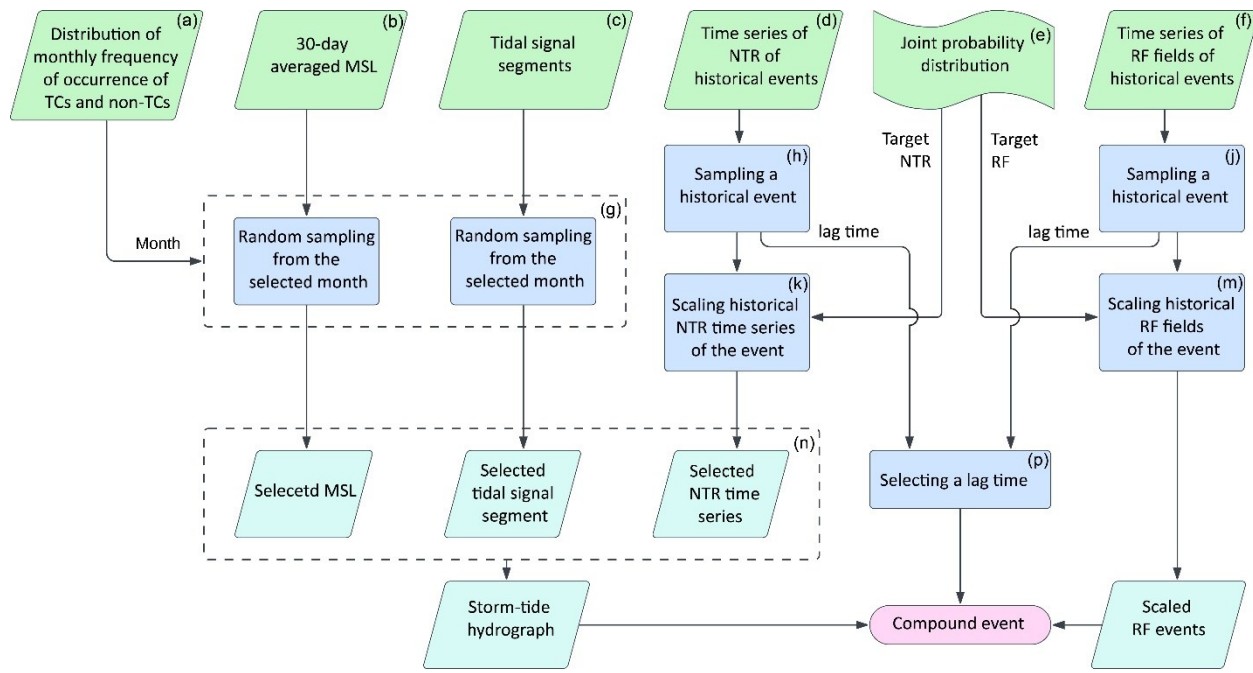

Figure 2: Workflow of the framework.

## 4.4 Assessing the effects of MSL and tidal variability on flood hazard

One advancement of the proposed framework over the approach outlined in Kim et al. (2023) is the inclusion of MSL and tides, along with their intra- and inter-annual variability. To assess how this variability affects compound flooding, we use the SFINCS model. SFINCS is a reduced-complexity model designed to simulate flooding from multiple drivers, such as storm surge, river discharge, and precipitation (Leijnse et al., 2021). It offers a simplified yet robust approach to modeling the complex interactions between flood drivers, balancing computational efficiency with accuracy. We define the flood model domain as the catchment area comprising the 14-digit hydrologic units of the two creeks (Newton and Little Timber Creeks) that surround Gloucester City to account for all the runoff that can produce pluvial flooding in the study site. The inland catchment area boundaries are defined as outflow boundaries to allow water to exit the domain. For the coastal boundary, we place an open boundary along the middle of the Delaware River, defined by the catchment polygons described earlier. We use the Coastal National Elevation Database (CoNED) from the U.S. Geological Survey, a Digital Elevation Model (DEM) with a horizontal resolution of 1 meter and a vertical accuracy of 10 cm (Danielson et al., 2016). We use the subgrid approach of SFINCS with a dual resolution of 10m and 1m. For surface roughness, we use land cover data from the NJDEP (New Jersey Department of Environmental Protection) Bureau of GIS, converting land classifications into Manning's coefficients based on guidance from the U.S. Army Corps of Engineers (US Army Corps of Engineers, 2024). Water level boundary conditions are provided as the time series at the location of the Philadelphia tide gauge. RF forcing is applied as spatially varying fields, with

the same resolution as the AORC data, and SFINCS interpolates these onto the model grid resolution. The model is run with the advection term neglected, solving the local inertia equations (we tested the sensitivity of the results when the advection term was enabled, but changes were negligible). We use the GPU version of SFINCS and ran the simulations on an Intel (R) Core (TM) i7-13700KF CPU and NVIDIA GeForce RTX 4080 GPU.

The lack of observed flood data to validate and calibrate flood models is a common challenge (see e.g., Merz et al., 2024; Molinari et al., 2019). For this case study, we search for historical flood information from several different sources, including high-water marks from USGS (United States Geological Survey), satellite images, the NOAA storm event dataset, FEMA Flood Risk Map, local news, and crowd-sourced platforms such as social media and citizen science platforms. However, very little information was found to perform a quantitative validation of the simulated water depths and extents. Due to the lack of observed historical flood data, we perform a qualitative validation comparing a few known flooded areas with simulated flooded sites for this qualitative validation, we also use local expert knowledge on areas that are frequently flooded as well as a few known flooded areas from past events from the previously listed sources. Overall, we find good agreement between the model output and the reported flood depths. A detailed description of the model validation can be found in Appendix 1 of Pollack et al. (2025).

To quantify the impact of including MSL and tide variations in the framework, we designed the following experiment. We use the most-likely event with 0.01 AEP (i.e., 100-year return period), determined from the derived joint probability distribution, as the target scenario for all simulations. Using the developed framework, we generate many most-likely 0.01 AEP events. A single event is then selected where the peak NTR coincides with high tide, as tidal variability would have less impact on flood depths if the peak NTR occurred during low tide. Then, we modify only the specific parameter of interest (MSL or tide) of the selected event while keeping all other event characteristics the same. To assess the impact of MSL, we change the MSL to the lowest and highest 30-day averaged MSL values recorded in the past five years and simulate the compound flooding. For tidal influences, we use tidal signal segments with the lowest and highest high tides over the last 18.6 years of the study period. This analysis allows us to assess the individual contributions from the variability of MSL and tides to overall flood hazard and better understand how critical it is to align with the seasonality when combining MSL and tide with NTR time series.

## 5 Results

### 5.1 Joint probability distribution

The threshold for NTR is set to 0.63 m, resulting in a total of 580 POT events (that is consistent with 5 events per year on average). For RF, thresholds are set to also obtain 580 POT events for each RF accumulation time from 1 to 48 hours. The 18-hour RF accumulation time exhibits the strongest correlation with the peak NTR. Therefore, the 18-hour RF accumulation is selected for subsequent analysis. After stratifying these events into TC and non-TC, 38 are identified as TCs when conditioned on NTR, and 43 when conditioned on RF, with the remaining events categorized as non-TCs. The conditioning variable of each stratified sample is fit to a Generalized Pareto Distribution (GPD). For the conditioned variable, several parametric

distributions are tested. Selected marginal distributions and quantile plots for each sample are shown in Fig. S1 in the
supplementary material. The rotated Tawn type 2 (180°) copula provides the best fit for both conditioning samples of TC
events. For the non-TC events, the Frank-Joe Copula is selected for the sample conditioning NTR and Clayton Copula for the
sample conditioning RF. The quantile isolines after combining the joint probability distributions of the two storm populations
(TC and non-TC) are shown in Fig. 3 (for a more detailed description, refer to Maduwantha et al. (2024)). Here we use the
framework to derive 5,000 combinations of peak NTR and RF by sampling from the fitted copulas such that the relative
proportion of extremes is consistent with the empirical distribution (see Fig. 3 (b)).

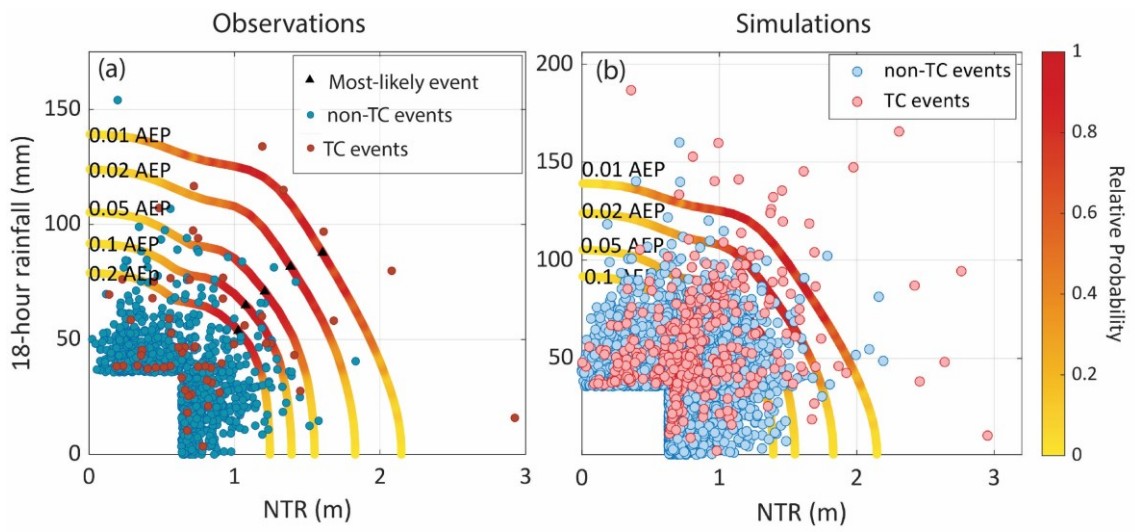

**Figure 3: Joint probability isolines after combining the AEPs of the two populations (TC and non-TC) with (a) observations, and (b) simulations. The color scale indicates the relative probability of events along the isolines. The location of the "most likely" event is assigned to the point with the highest relative probability density on an isoline (black triangles in (a))**

**5.2 Characteristics of TC events and non-TC events**

We use Kendall's rank correlation coefficient $\tau$ to measure the strength of dependence between different attributes of observed
events falling into the TC and non-TC categories. The correlations between NTR duration and peak NTR (Fig. 4 (a)), NTR
intensity and peak NTR (Fig. 4 (d)), total RF and peak hourly RF (Fig. 4 (k)) are strong, positive, and statistically significant.
The lag times of the observed events are predominantly positive, indicating that the peak hourly RF typically occurs before
the peak NTR. The correlation between lag time and peak RF (Fig. 4 (i)) is weakly to moderately negative, but statistically
significant only for the non-TC sample. However, Fig. 4 (e) and Fig. 4 (i) show that events with higher peaks of NTR or RF
generally tend to have shorter lag times. There is no significant correlation between RF duration and NTR duration in both TC
and non-TC samples (see Fig 4 (c)). To further examine differences in the pairwise correlations in TC and non-TC samples,
we derive the confidence intervals associated with the values of Kendall's $\tau$ (Fig. 5). Only the NTR hourly peak vs. RF hourly

peak and NTR intensity vs. total RF exhibit non-overlapping 95% confidence intervals, whereas in all other cases, the confidence intervals for TC and non-TC events overlap.

The NTR duration, NTR intensity, lag time, RF duration, peak hourly RF, and total RF observations are fitted to various parametric distributions, with the best fitting selected based on AIC. Fig. 6 displays the estimated parameters of the selected distributions along with their 95% confidence intervals. For all parameter values, the confidence intervals for TC events overlap

with those of non-TC events, except for the scale parameter of the RF duration. The goodness of fit of the parametric distributions is shown in Fig. S2 of the supplementary material. As described in Section 4.2, we also check the time evolution of the NTR and basin-averaged RF of the observed POT events. Fig. 7 shows the hourly time series of NTR and basin average RF of observed events around the peak. Although peak RF is higher for TC events compared to non-TCs, the overall shape of the NTR time series and basin-average RF is similar for both storm types. Therefore, TC and non-TC RF and NTR time series

are randomly sampled (and scaled to the target peak values) without stratifying by storm type.

We emphasize that stratification is still conducted and important when deriving the joint probability distribution because TC and non-TC events exhibit different dependence between NTR and RF. However, the relevant characteristics of the complete time series of the different event types are similar, as shown in this section. Therefore, to have a larger sample to draw from (especially in the TC case) we do not treat TC and non-TC events separately when selecting observed event time series for

subsequent scaling. We elaborate on this more in the Discussion section.


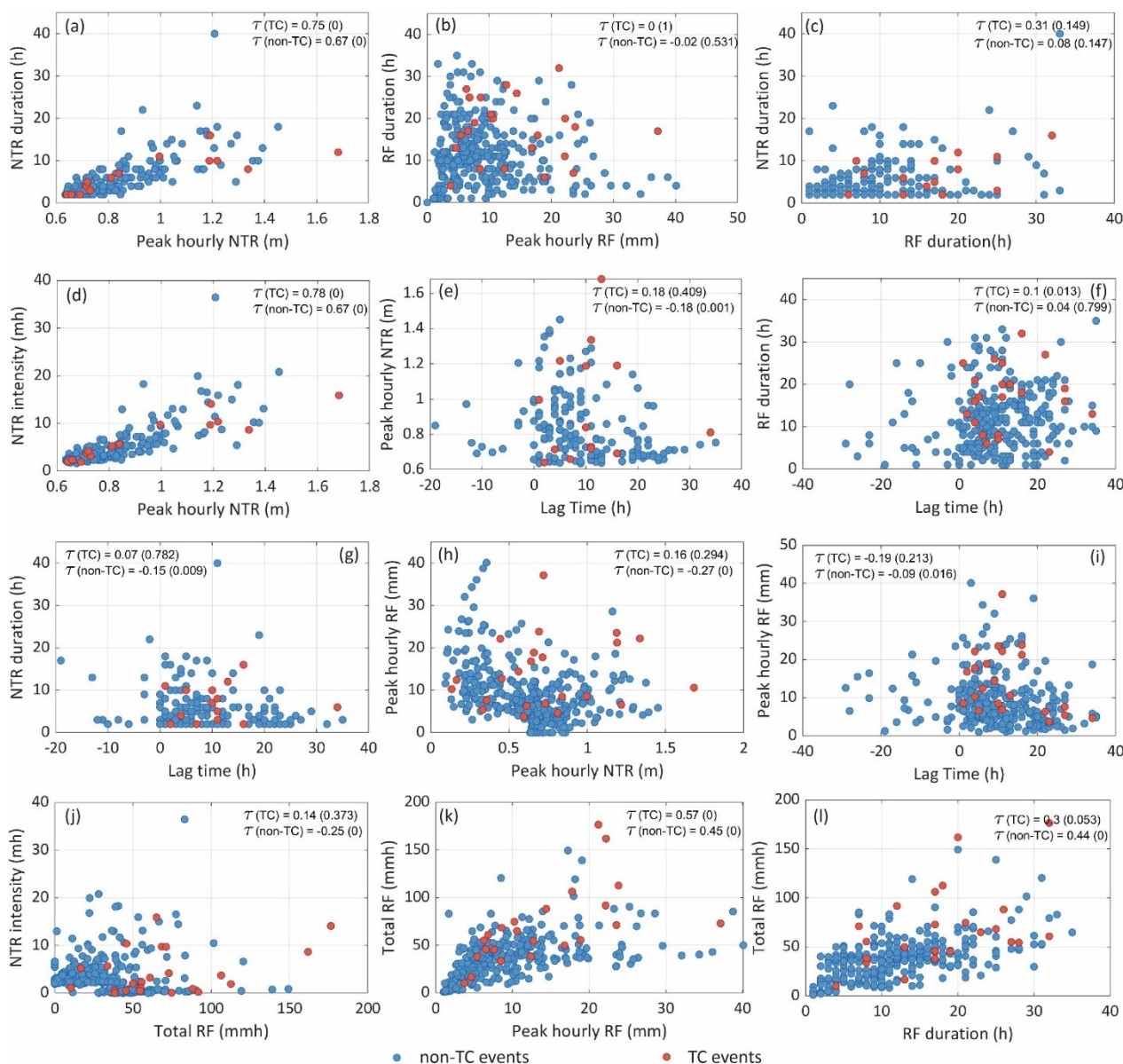

Figure 4: Scatter plots between (a) NTR duration and peak NTR, (b) RF duration and peak hourly RF, (c) NTR duration and RF duration, (d) NTR intensity and peak NTR, (e) peak NTR and lag time, (f) RF duration and lag time, (g) NTR duration and lag time, (h) peak hourly RF and peak NTR, (i) Peak hourly RF and lag time, (j) NTR intensity and total RF, (k) total RF (sum of all the basin-averaged hourly RF quantities of the event) and peak hourly RF, of observed TC events (red) and non-TC events (blue). Kendall's τ for each sample with the corresponding p-value (in brackets) is shown in each panel.


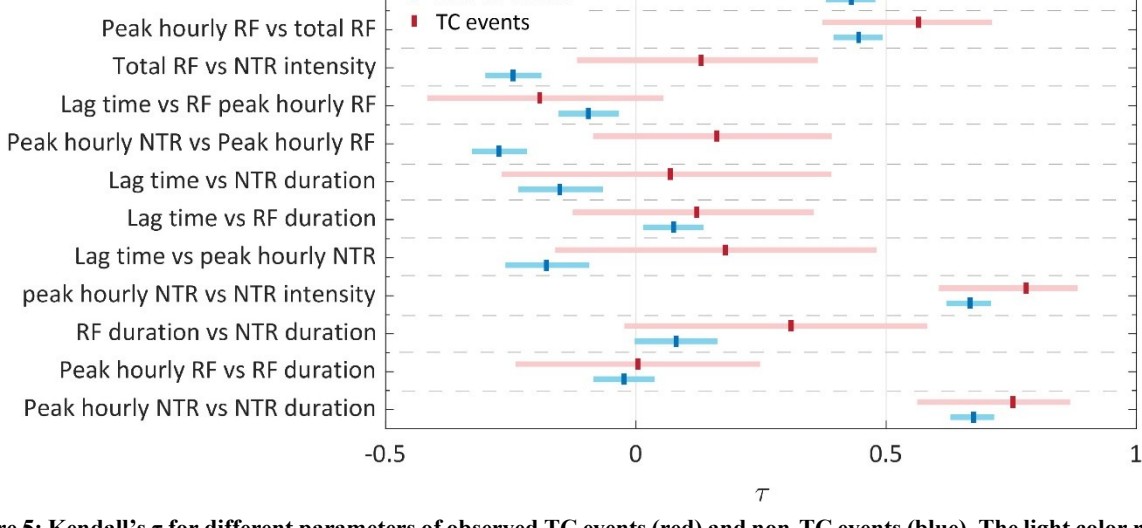

**Figure 5: Kendall's τ for different parameters of observed TC events (red) and non-TC events (blue). The light color range indicates the associated 95% confidence intervals.**


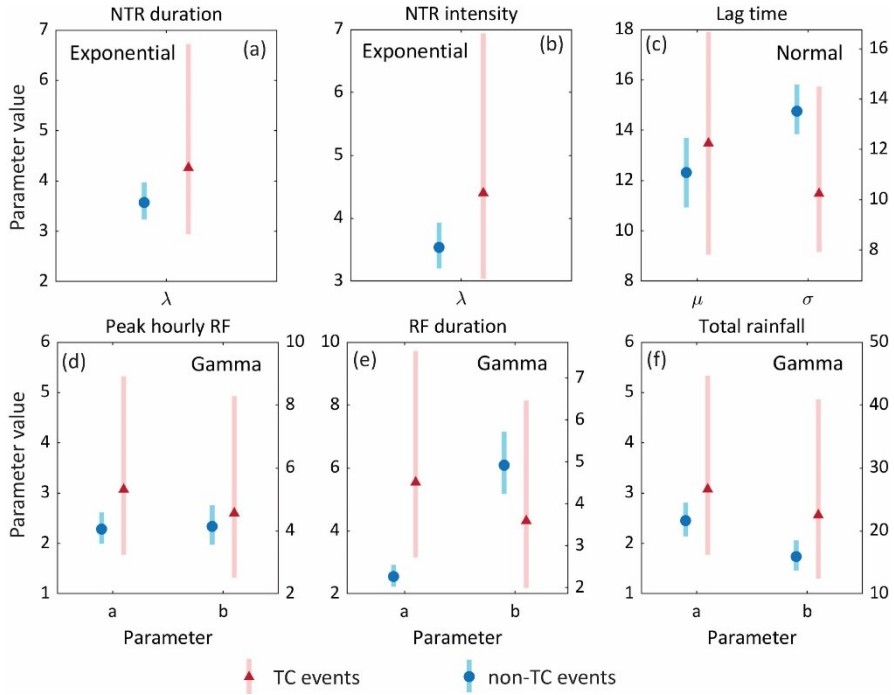

**Figure 6: Parameter values of the fitted parametric distributions with their 95% confidence intervals for TC events (red) and non-TC events (blue). The selected parametric distribution is shown in each panel.**

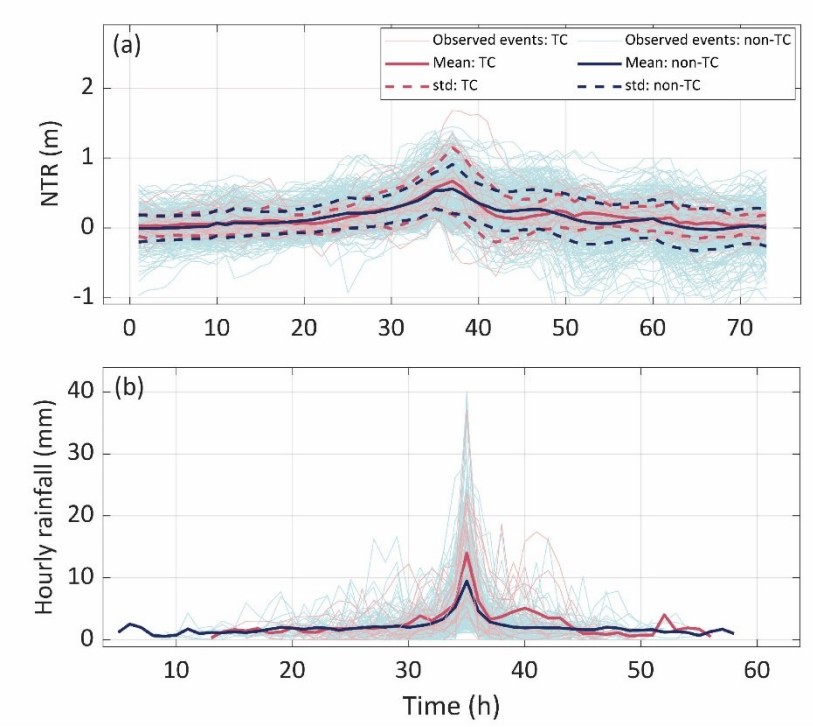

**Figure 7: Houry time series of (a) NTR and (b) basin average RF of observed events besides the peak. The solid lines show the mean value of each time step of TC events (red) and non-TC events (blue). The dashed lines of (a) represent the standard deviation around the mean at each time step.**

## 5.3 Event generation process

Fig. 8 illustrates the procedure for generating an event with a 106-year joint return period, consisting of a 1.75 m NTR and 80 mm 18-hour basin-average RF. Since the target event was simulated from the copula that was fit to the TC sample, the event month was randomly sampled from the frequency of TC occurrences in each month (Fig. 8 (d)). For the selected event, the month of July was sampled. After that, a MSL value of 0.3 m was selected from the MSL distribution for the month of July (Fig. 8 (e)). To generate the storm tide hydrograph, an NTR time series was sampled from the observed events (regardless of storm type) (Fig.8 (g)) and scaled to match the target value (Fig. 8 (h)). The NTR time series was subsequently combined with the sampled MSL and a randomly selected tidal signal segment, chosen from the set of tidal signal segments for the month of July (Fig. 8 (f)). For generating RF fields, an RF event was sampled (regardless of storm type) from all available events (Fig. 8 (b)) and scaled to match the target 18-hour RF (Fig. 8 (c)). Fig. 8 (k) shows the scaled RF fields at selected hours, demonstrating the spatio-temporal variability in the RF fields. A 6-hour time lag, originally associated with the selected RF event, was used to combine the RF time series with the storm tide hydrograph (Fig. 8 (m)).

The proposed framework was implemented to generate 5,000 synthetic events, consisting of hourly still water levels (storm tide hydrograph) at the Philadelphia tide gauge and hourly RF fields over the Gloucester City catchment. Fig. 9 shows the scatter plots comparing various characteristics of the time series, including hourly peaks, durations, intensities, and lag times for both observed and simulated events. Overall, the spread and correlation for each pair of parameters in the simulated events are consistent with those in the observed events.

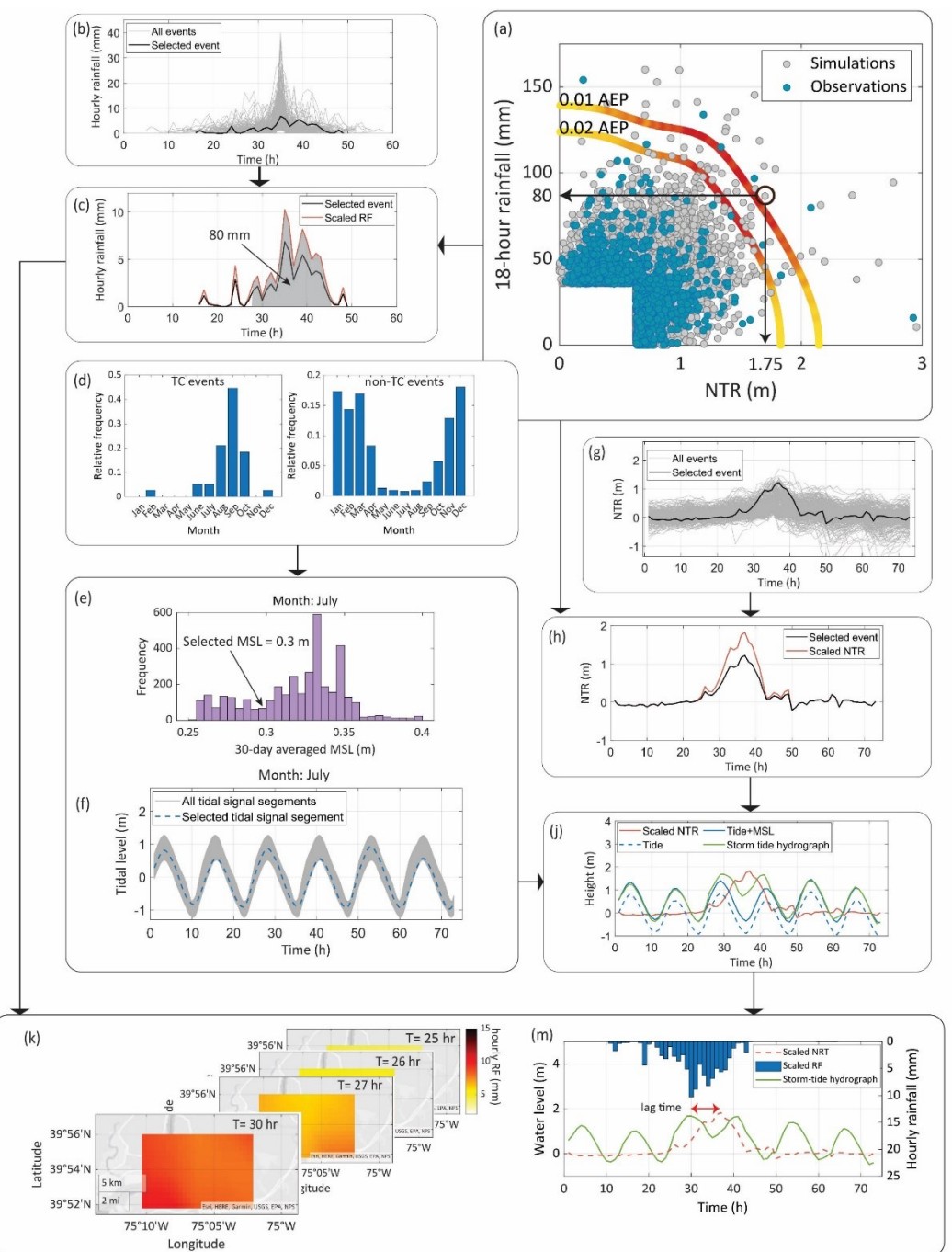

Figure 8: The demonstration of the event generation process using an example target event with 1.75 m NTR and 80 mm 18-hr RF. Panels: (a) Joint probability distribution, (b) Observed RF time series, (c) Selected and scaled RF time series, (d) Monthly frequency of occurrence of TC events and non-TC events, (e) MSL distribution of the month July, (f) Tidal signal segments of the month July, (g) Observed NTR time series, (h) Selected and scaled NTR time series, (j) Storm tide hydrograph, (k) Scaled hourly RF fields over the Gloucester City catchment, (m) Synthetic compound event comprised of storm tide hydrograph (including scaled NTR) and scaled RF.

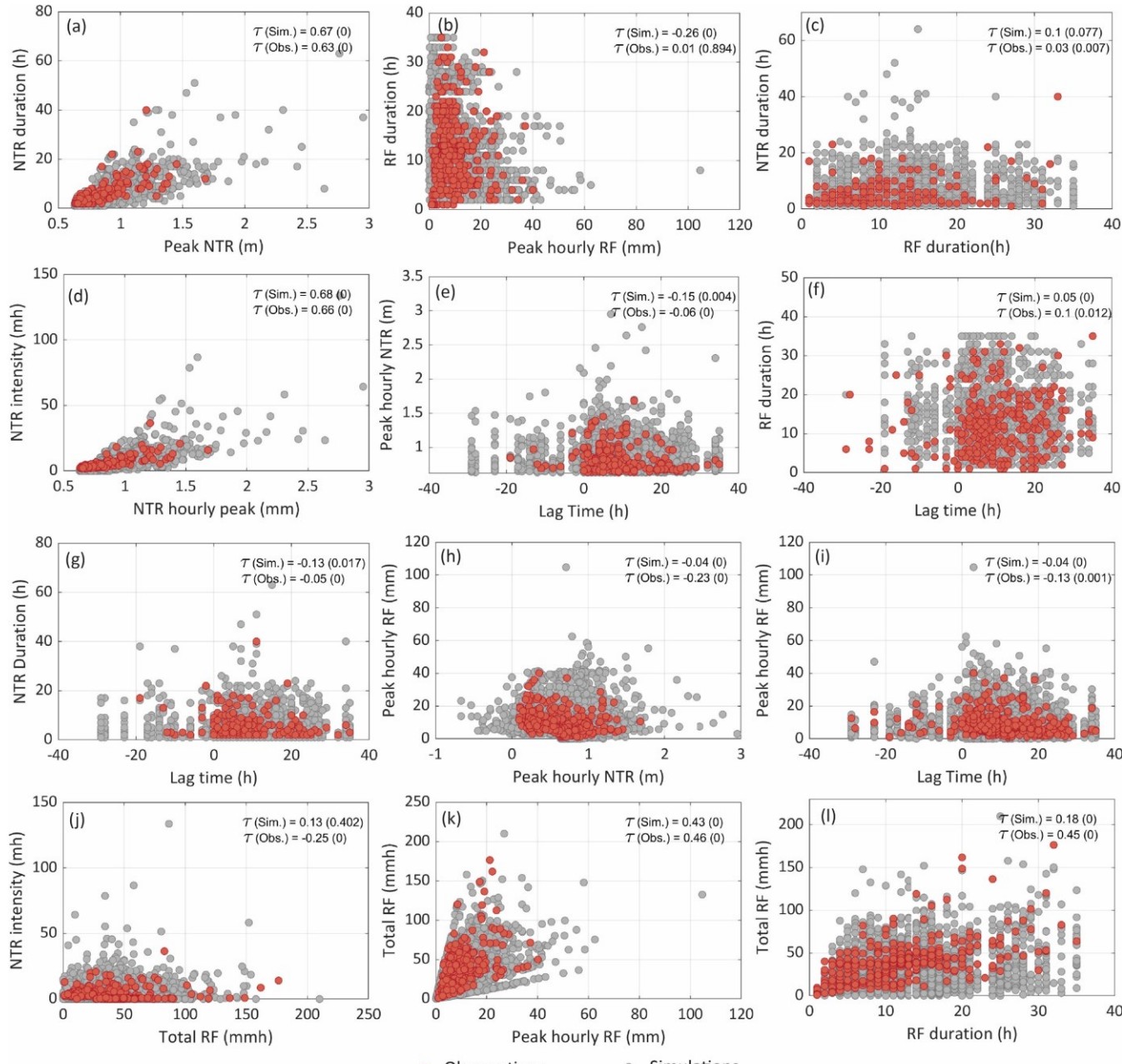

**Figure 9: Scatter plots between (a) NTR duration and peak NTR, (b) RF duration and peak hourly RF, (c) NTR duration and RF duration, (d) NTR intensity and peak NTR, (e) peak NTR and lag time, (f) RF duration and lag time, (g) NTR duration and lag time, (h) peak hourly RF and peak NTR, (i) Peak hourly RF and lag time, (j) NTR intensity and total RF, (k) total RF (sum of all the basin-averaged hourly RF quantities of the event) and peak hourly RF, of observed events (red) and simulated events (gray). Kendall's τ for each sample with the corresponding p-value (in brackets) is shown in each panel.**


**5.4 Flood model simulations and role of MSL and tide variation**

A most-likely 0.01 AEP event (see black triangle in Fig. 3 (a)) was used to assess the impact of tidal and MSL variability on
flood depth and extent. To assess MSL impact, we simulate flooding by adjusting the MSL to the lowest (-0.198 m, above
NAVD88) and highest (0.540 m, above NAVD88) 30-day average values recorded over the past five years. For tidal influences,
we use tidal segments with the lowest and highest high tides observed over the last 18.6 years. Fig. 10 illustrates the maximum
flood depth and extent resulting from each scenario during the flood model simulations. There is a significant difference in
flood depth and extent when comparing the simulation results of applying the maximum and minimum tide (or MSL). Flood
depths reach up to 1.5 meters in certain areas when the highest 30-day MSL is used for generating the storm-tide hydrograph.
The difference in flood depths between using the highest and lowest 30-day MSL reaches up to 1 m in some areas of the city.
Similarly, applying the tidal signal segment with the highest high tide causes flood depths to reach 2 m in several areas, with
increases over 1.2 m compared to using the segment with the lowest high tide. These changes in flood depths are particularly
pronounced along the Delaware River and Newton Creek, where the influence of coastal water levels is strongest.

**6 Discussion**

A detailed description of the procedure for estimating the joint probability distribution applied in this study is provided in
Maduwantha et al. (2024). When applying the two-way sampling to extract POT events, we used a 3-day pairing window to
capture peak NTR and RF, following similar studies (Couasnon et al., 2020; Kim et al., 2023). We also manually checked the
RF and NTR time series of POT events and found that a 3-day window was generally sufficient to capture both peaks in the
vast majority of cases. To ensure independence within the POT samples, previous studies have applied various declustering
windows (e.g., 3 days (Haigh et al., 2016), 7 days (Santos et al., 2021), 10 days (Kim et al., 2023), and 14 days (Terlinden-
Ruhl et al., 2025)). Longer declustering windows are often adopted when the influence of river discharge is present, as its
effects can persist for several days or more (Terlinden-Ruhl et al., 2025). In this study, we use a 5-day declustering window
(2.5 days before and after the event peaks), as highly elevated NTR rarely lasts more than 5 days at the tide gauge location.
Previous studies have applied various search radii to identify TC events, such as ~400 km (Kim et al., 2023) and 500 km
(Towey et al., 2022). In this study, we tested the sensitivity of the correlation between peak NTR and peak accumulated RF to
the TC search radius, following Kim et al. (2023). Increasing the search radius captures more nearby TC tracks but also
introduces events that are too distant to strongly influence flooding drivers at the study site, thereby reducing the overall
correlation between RF and NTR of the TC sample. We selected a 350 km search radius, as it provided a higher correlation
between drivers while still retaining a reasonable number of TC events in the sample.
Maduwantha et al. (2024) identified a strong correlation between peak NTR and peak RF when the extreme events are caused
by TCs in the Gloucester City region, suggesting that there is a higher potential for compound flooding by TCs in the study
region (Fig. S3 (a) and (c) in supplementary material). The non-TC events, which include ETCs and convective RF events,
exhibit a weaker correlation between peak NTR and RF. Consequently, TC and non-TC events were treated as two distinct

populations in the joint probability analysis, leading to more accurate and robust estimates compared to modeling them as a single population (Maduwantha et al., 2024). The joint probability distributions of peak NTR and peak RF of TC and non-TC storms are substantially different. Small to moderate compound events are more frequent in the non-TC storms, whereas the most extreme compound events are more likely generated by TCs. For example, the selected event for demonstrating the event-generation framework (NTR = 1.75 m, 18-h RF = 80 mm) corresponds to a ~106-yr joint return period in the combined joint probability distribution. The same event has a joint return period of ~111 yrs in the TC sample and ~2431 yrs in the non-TC sample, highlighting that rare events are primarily associated with TCs." Here, the generated 5,000 combinations of peak NTR and RF by sampling from the fitted copulas provide 1,000 years' worth of extreme events (5 events per year on average), reflecting the joint probability distribution of NTR and RF.

Considering the distinct properties of TCs compared to ETCs and other storm types, it is crucial to account for the unique characteristics of these flood drivers in the synthetic event generation process. Therefore, the most effective approach would be to use observed time series of flood drivers from TC events exclusively for generating synthetic TC events, while using those from non-TC events separately to generate synthetic non-TC events. This separation allows for a more accurate representation of the differences in timing (of peak storm surge and peak RF), intensity, duration, and spatial patterns between TC and non-TC events, ensuring that the synthetic events realistically reflect the distinct physical properties associated with each storm type. However, the small number of TCs in the historical record, due to their infrequent occurrence, presents a challenge when generating many synthetic events. A limited TC dataset may not fully reflect the

inherent variability and the full range of possible events through the event generation process. Therefore, we assess whether the event generation process can be applied to the entire sample combining both TC and non-TC events while still preserving key characteristics of the flood drivers. To inform this decision, we examined various time series attributes of NTR and RF, such as magnitudes, durations, shapes, and timing.

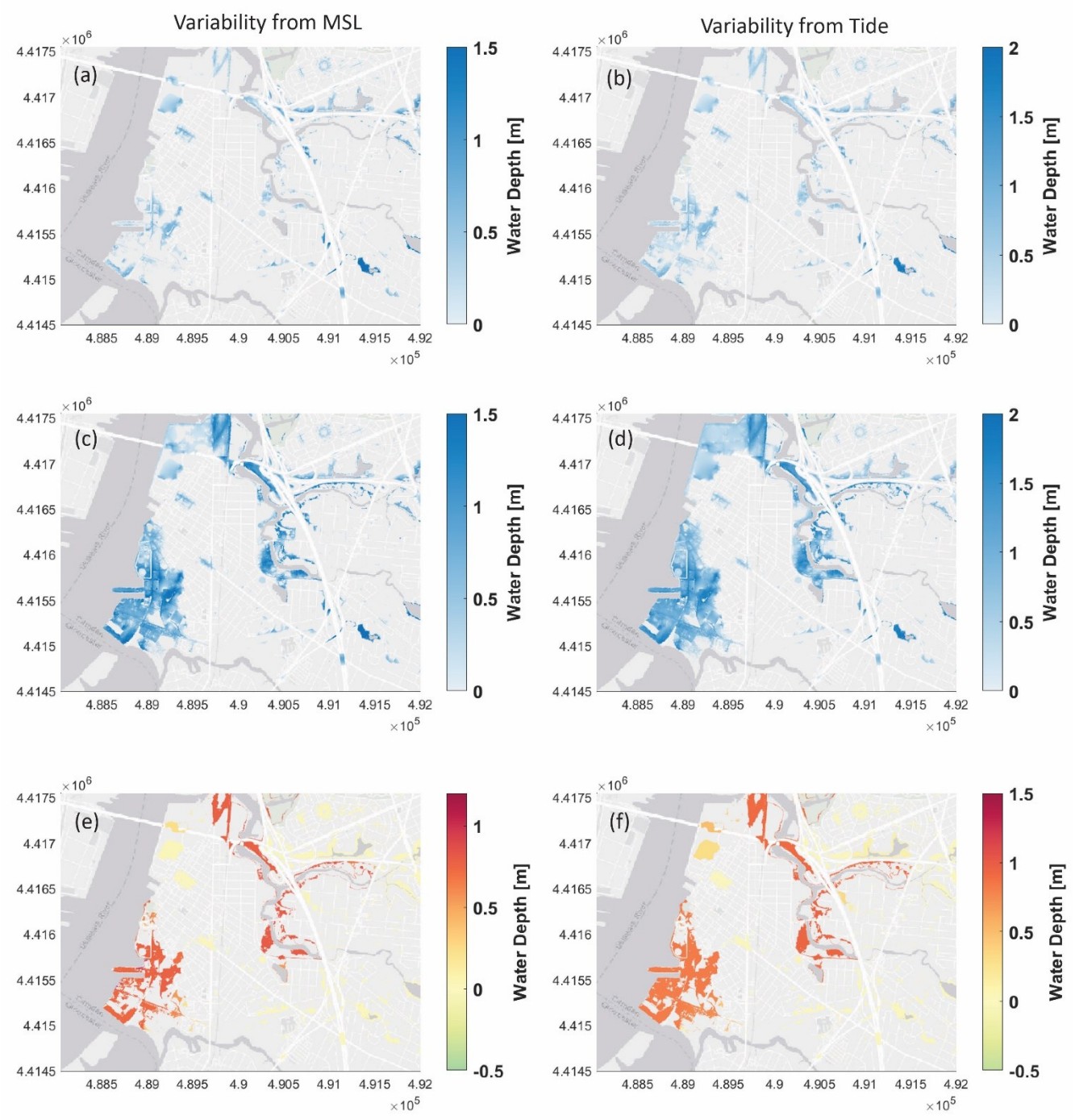

**Figure 10: Changes in flood depths associated with variability in MSL (left) and Tide (right) of a selected 0.01 AEP most-likely event. (a) When considering the lowest 30-day MSL, (b) when considering the tidal segment with the lowest high tide, (c) when considering the highest 30-day MSL, (d) when considering the tidal segment with the highest high tide, (e) the difference between (a) and (c), (f) difference between (b) and (d). (X, Y coordinates system: UTM- zone 18N).**


Kendall's τ for the hourly peak and duration of the NTR time series shows a strong positive correlation (see Fig. 4 (a)), suggesting that more intense storm surge events tend to last longer in the study region. In contrast, for RF there is no significant correlation between the hourly peak and the duration of the basin-average RF, as indicated by τ values closer to zero (see Fig. 4 (b)). However, the confidence intervals of τ, as shown in Fig. 5, indicate that the strength of dependence between the tested

characteristics does not differ significantly between the two storm types. It is well known that TCs typically produce more intense RF than ETCs, whereas ETCs tend to be larger in size and generate RF for prolonged durations (e.g., Orton et al., 2016; Sinclair et al., 2020). However, this behavior is not evident in the statistical properties of the observed POT events, as shown in Figs. 4 to 6. Several factors could explain this. First, the non-TC sample may include TCs that passed beyond the 350 km search radius but still contributed RF and storm surge to the Gloucester City catchment. Second, the non-TC sample also

contains locally generated convective RF events, which, although shorter in duration, can produce severe RF intensities (Pfahl and Wernli, 2012). Further, the smaller number of observed TC events may lead to statistically significant correlations going undetected and wider confidence intervals, limiting the ability to discern distinct patterns. One option to overcome the limited TC sample size in our analysis is employing physics-based models to generate time series of flood drivers from synthetic TC tracks (e.g., Emanuel et al., 2006; Gori et al., 2020). We plan to explore this in future work. The relatively small size of the

Gloucester City catchment also means that we analyze only parts of the spatial variability associated with different storm types. The comparison of distribution parameters fitted to peak RF, total RF, RF duration, lag time, NTR duration, and NTR intensity also suggests no significant differences between the various characteristics of TCs and non-TCs (see Fig. 6). Similarly, the shapes of the NTR and basin-average RF time series produced by TCs are not significantly different from those generated by non-TC events (see Fig. 7). Given these results we conclude that sampling and scaling of the observed event time series (i.e.,

water level hydrographs and RF hyetographs) separately for the two storm types would produce similar results compared to the ones we derive without stratifying. Note, that stratification is still applied when deriving the joint probability distribution since the dependence structure of the peaks of NTR and RF is substantially different. Importantly, this applies to the specific study location. In other places, significant differences may exist in the time series characteristics between TC and non-TC samples (as discussed in Section 4.2), warranting that the event generation process is also conducted separately for each storm

type.

In the event generation process described in Section 4.3, steps are taken to ensure the synthetic events are both realistic and physically plausible. While lag times between peak NTR and peak RF can vary a lot, more extreme events tend to exhibit shorter lag times (see Figs. 4 (e) and 4 (i)). To incorporate this behavior into the synthetic events, we not only select nearby historical events for scaling but also adopt the lag time from one of the selected events. At the Philadelphia tide gauge, peak

NTR often occurs 4–5 hours before the next high tide (see Fig. S4 in the supplementary material). To account for this, we combined scaled NTR with tide predictions using the observed lag between peak NTR and subsequent high tide of the sampled NTR event (see Section 4.3.3). These steps ensure that synthetic compound events retain the same temporal dynamics as similar observed events.

MSL exhibits both long-term trends and seasonal variability, which is often driven by regional climate characteristics (Barroso et al., 2024). Detection of this seasonality is crucial, as the risk of flooding increases significantly when elevated MSL coincides with storm activity and/or seasonal high tides (known as king tides), compared to when these peaks are out of phase (Barroso et al., 2024; Dangendorf et al., 2013; Thompson et al., 2021). At the Philadelphia tide gauge, the 30-day averaged MSL varies by approximately 0.7 m over the last five years of the study period, highlighting the importance of incorporating this variability into flood modeling frameworks. The long-term variations of tides have also been linked to increases in high-tides and extreme coastal flooding (Enríquez et al., 2022; Thompson et al., 2021). These tidal variations arise from the nodal and perigean modulations, with cycles of 18.6 and 4.4 years respectively. To account for these tidal variations, we use 3-day tidal signal segments over the most recent 18.6 years of the study period to generate the synthetic storm events. The framework was applied to generate 5,000 synthetic events, and the comparisons of scatter plots in Fig. 9 indicate that the characteristics of the simulated events, such as hourly peaks, durations, intensities, and lag times are consistent with the observed events.

The results of the compound flood model simulations show that a substantial portion of the study area is impacted by a 0.01 AEP compound flood event. Still, the flood depth varies significantly depending on the MSL and tidal conditions (see Fig. 10). An event with 0.01 AEP (i.e., joint probability between peak RF and peak NTR) can produce up to 1 m difference in flood depth depending on MSL conditions, while the prevailing tidal conditions can lead to differences of up to 1.2 m. These changes are particularly evident in areas along the Delaware River and Newton Creek, where the influence of coastal water levels is the largest. It is important to note that these variabilities are solely due to the influence of MSL and tides, and do not account for additional variability from different combinations of NTR and RF peaks along the 0.01 AEP isoline or other factors (Jane et al., 2022). Nonetheless, the substantial differences in flood depths highlight the critical importance of accurately representing MSL and tidal conditions, which we achieve in the proposed framework by randomly sampling from their monthly distributions. Analyzing only the most likely event, even if it appears to be the most plausible based on observations, does not capture the range of flood levels that could be generated by different combinations of flood drivers (i.e., NTR and RF) with different time series properties. Therefore, the flood model simulations presented here are aimed at evaluating the importance of explicitly accounting for the variability of MSL and tides, and not to produce comprehensive probabilistic flood maps. In a separate study (Santamaria et al., 2025), we simulated flooding of 5,000 synthetic storms at this site and found large variation in resultant flooding, even for events with similar joint return periods. However, attributing this variability to a single factor like MSL or tides is challenging due to the complexity of their interactions.

One key assumption of the framework is that uniform scaling (also referred to as "same frequency amplification") of flooding-driver time series creates a realistic compound event. This approach has been widely adopted in previous studies to construct design hydrographs and hyetographs (e.g., Serafin et al., 2019; Moftakhari et al., 2019; Zellou and Rahali, 2019; Kim et al., 2023; Liu et al., 2024; Xu et al., 2024). However, assessing whether each generated synthetic storm event is physically realistic is challenging. Ideally, a direct one-to-one validation against observed events (verifying whether every synthetic event has a similar observed event) would provide the most rigorous test. Yet such validation is impossible given the limited availability of observations, which is why the synthetic event generation is necessary in the first place. Instead, we tested the framework

by comparing statistical properties of key time series characteristics between observations and synthetic events (Fig. 9). Another limitation of the proposed framework is that certain characteristics of synthetic events, such as RF duration and lag times, are limited to the observed values. To generate more diverse lag times, the observed lag times could be fitted to a parametric distribution (or alternatively to a copula that accounts for the dependence between peak values and lag times) and then one could sample lag times from the fitted distribution during the event generation process. This would introduce unobserved lag times into the synthetic events, enhancing their diversity. Additionally, the stratification of POT events utilizes a simple yet commonly used approach (e.g., Kim et al., 2023; Maduwantha et al., 2024) as discussed in Section 4.1. However, this method may not capture all TCs, particularly those that produce significant RF and storm surges from distances greater than 350 km. Such events are classified as non-TC events here, meaning the analysis in Section 4.2 may not fully reflect the true characteristics of TC and non-TC events.

Although measures are taken to prevent the generation of physically unrealistic events (see Section 4.3), it cannot be fully ruled out. For instance, when generating many peak NTR-RF combinations from the multivariate statistical model, unbounded marginal distributions can produce implausible extreme events that would result in unrealistic flood depths for those particular events. How much that affects the overall results depends on the type of analysis and how the flood information from individual synthetic events is used. Implementing a quality control process, e.g., using probable maximum precipitation or existing data on maximum storm surge potential (in the U.S. such data is available from a large number of SLOSH simulations) could help filter out such unrealistic events, ensuring that the resulting synthetic event set remains feasible for a comprehensive flood risk assessment.

**7 Conclusions**

This paper presents a novel framework for generating synthetic events consisting of RF fields and (coastal/estuarine) water level time series, which can serve as boundary conditions for compound flood models. The framework explicitly accounts for different storm types in estimating the joint distribution of flood drivers and derives a large sample of peak NTR-RF combinations. Historic time series are scaled to match the target peaks, with the observed events chosen to ensure that the re-scaled events are physically plausible. We applied this framework to Gloucester City in New Jersey, a coastal city that is exposed to flooding from multiple water sources and storm types. The results demonstrate that the simulated events are consistent with observed events while covering unobserved portions of the event space. Results of the flood modeling indicate that substantial variability in flood depth can arise solely from different MSL and tidal conditions, even when peak NTR and RF values are the same. This emphasizes the importance of accounting for the variability in time series dynamics, MSL, and tidal conditions in compound flood risk assessments. While we focus on historical observed events, the framework can be used with model output data including hindcasts or future projections.

**Code availability**

The marginal distribution fitting and copula selection were done using the MultiHazard R package, which can be downloaded from GitHub at https://github.com/rjaneUCF/MultiHazard (DOI: https://doi.org/10.5194/nhess-20-2681-2020.). The other codes are available on GitHub at https://github.com/CoRE-Lab-UCF/MACH-Compound-Flooding. (The DOI and the final version of the codes will be available after addressing the reviewers' comments and suggestions.)

**Data availability**

The hourly water level data at the Philadelphia tide gauge (St. ID: 8545240, St. ID: 8545530) can be accessed through the National Oceanic and Atmospheric Administration (NOAA: http://tidesandcurrents.noaa.gov/). The HURDAT2 data are available from https://www.nhc.noaa.gov/data/hurdat. The measured rainfall data used in this paper can be downloaded through NOAA's National Climatic Data Center's (NCDC) archive of global historical weather and climate data at https://www.ncdc.noaa.gov/cdo-web. The AORC (4-km) Version 1.1 datasets can be obtained from the NOAA and available at https://hydrology.nws.noaa.gov/aorc-historic/.

**Author contribution**

The study was conceived by TW and PM. PM developed the methodology, undertook the analysis, and wrote the first draft of the paper under the guidance of TW, SSA, RJ, and SD. HK and GV contributed to technical discussions in the early stages of the analysis. All authors co-wrote the final paper.

**Competing interests**

The authors declare that they have no competing interests.

**Acknowledgment**

PM, TW, SSA, and SD were supported by the National Science Foundation as part of the Megalopolitan Coastal Transformation Hub (MACH) under NSF award ICER-2103754. MACH contribution no. 68-P.

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
