# Peer review of "Generating Boundary Conditions for Compound Flood Modeling in a Probabilistic Framework"

_EGUsphere, 2025_

## Author Comment (AC1)

This manuscript presents a modeling framework for evaluating the joint influence of non-tidal residuals (NTR), rainfall (RF), and mean sea level variability on coastal flooding in the Gloucester City area, using the SFINCS hydrodynamic model and a copula-based statistical approach. The topic is timely and relevant, and the study is generally well-structured with a strong emphasis on scenario-based risk quantification. However, several methodological choices—particularly regarding data selection, parameter thresholds, and model assumptions—require further clarification or justification. Issues such as the generalization of AORC performance, the treatment of tropical versus non-tropical events, and simplifications in the SFINCS physics raise concerns about the robustness and generalizability of the findings. Despite these limitations, the study offers valuable insights into compound flood risk assessment. Detailed comments are provided below, which I hope will be useful in clarifying and strengthening the manuscript:

We thank the reviewer for their constructive and insightful comments, which have helped enhance the overall quality of our manuscript. Below, we provide detailed responses to each point and outline how we plan to address them in the revised manuscript. The changes to the existing text are highlighted using track changes with line numbers in the original manuscript.

Page 5, Line 146: The sentence claiming that AORC has "higher accuracy" than other gridded rainfall datasets seems too general. For example, radar-based products like MRMS have been shown to perform as well as or better than AORC in some events, including Hurricane Harvey (e.g., Gao et al., 2021; Gomez et al., 2024). I suggest the authors either include MRMS in their comparison or rephrase the sentence to clarify that AORC's performance advantage may depend on the region or event type.

We thank the reviewer for bringing up this point and agree that MRMS products have demonstrated higher accuracy than many other datasets, while offering higher resolution (1 km and hourly for historical). However, our primary reason for selecting AORC over MRMS rainfall data is its longer temporal coverage (1979 to present), which allows the extraction of rainfall fields for a sufficiently large number of observed events required in our event-generation process. In contrast, MRMS precipitation data are only available from 2012 onwards (NOAA/NSSL, 2023), which limits their applicability for our framework.

We will revise the relevant section in the original manuscript as follows:

L135 : "We use both gridded RF data from the Analysis of Period of Record for Calibration (AORC) from 1979 to 2021 and hourly RF gauge data at the Philadelphia International Airport from 1900 to 2021 (Kitzmiller et al., 2018). Although radar-based quantitative precipitation estimates, such as the Multi-Radar Multi-Sensor (MRMS) products, often provide higher accuracy compared to other gridded rainfall products, their temporal coverage is relatively short (Gao et al., 2021; Gomez et al., 2024). We use AORC rainfall data because of its availability from 1979 onward and its demonstrated higher accuracy among products with similar temporal coverage, while offering hourly data with ~4 km spatial resolution (e.g., Hong et al., 2024; Kim and Villarini, 2022) AORC RF data has demonstrated higher accuracy compared to other gridded data sets while offering an hourly temporal resolution and ~4 km spatial resolution (e.g., Hong et al., 2024; Kim and Villarini, 2022). To leverage the long-term in-situ observations and obtain more robust results from the statistical analysis, we apply a bias correction to the hourly RF gauge data, to match with the hourly basin-average RF values calculated from AORC. The bias correction is performed using the

quantile mapping method, fitting both the hourly measured gauge data and the hourly AORC basin-average data to gamma distributions (for more details see Maduwantha et al. (2024))."

Additional Ref.

NOAA/National Severe Storms Laboratory. (2023). Multi-Radar Multi-Sensor (MRMS) system. National Oceanic and Atmospheric Administration. https://www.nssl.noaa.gov/projects/mrms/

Page 6, Line 160-166: In Section 4.1, several choices such as the 3-day pairing window for NTR and RF, the 5-day declustering period, and the 350 km radius for identifying TC events are not clearly explained. It would be helpful to clarify whether these are based on physical reasoning, prior studies, or simply assumptions made for this analysis. Providing brief justifications or references would improve transparency and reproducibility.

Both the choices of peak pairing window and declustering windows depend on the location-specific storm climatology and hydrologic response of the catchment. The choice of a 3-day pairing window for NTR and RF was based on the typical timescales over which storm surge (here represented by NTR) and rainfall peaks can occur in a compound flood event. We manually checked the time series of RF and NTR of POT (peaks over threshold) events and found that a 3-day pairing window was sufficient to capture both NTR and RF peaks of most events. Using a longer window could capture peaks from two different storm events and treat them as a single event in the bivariate frequency analysis. Furthermore, a 3-day pairing window has been found to be appropriate in similar previous studies (Couasnon et al., 2020; Kim et al., 2023; Maduwantha et al., 2024).

Different declustering windows have been used in previous studies to assure sampled extreme events are independent (e.g., 3 days (Haigh et al., 2016), 7 days (Santos et al., 2021), 10 days (Kim et al., 2023), and 14 days (Terlinden-Ruhl et al., 2025). Longer declustering windows are often adopted when the influence of river discharge is present, as its effects can last several days (Terlinden-Ruhl et al., 2025). Considering the duration of the storm events at our tide gauge location of interest, we use a 5-day window, since elevated NTR rarely lasted more than 5 days.

Previous studies have applied various search radii to identify TC events using a similar approach to ours (e.g., ~400 km (Kim et al., 2023) and 500 km (Towey et al., 2022)). In this study, we tested the sensitivity of the correlation between peak NTR and peak accumulated rainfall to the TC search radius, following Kim et al. (2023). Increasing the search radius captures more nearby TC tracks but also introduces events that are too distant to influence flood drivers at the study site, thereby reducing the overall correlation between rainfall and NTR of the TC sample. This is also influenced by TC intensity; for example, a Category 5 hurricane is more likely to generate significant storm surge even when passing far from the tide gauge, compared to a Category 1 hurricane. In our analysis, we considered all cyclones listed in the HURDAT2 database as TC events, regardless of their sustained wind speed or whether they were hybrid systems during part of their lifetime. We therefore selected a 350 km search radius, as it provided a higher correlation between drivers while still retaining a reasonable number of TC events in the sample.

To support these choices in the main manuscript, we add the following paragraph:

L 415: "A detailed description of the procedure for estimating the joint probability distribution applied in this study is provided in Maduwantha et al. (2024). When applying the two-way sampling to extract POT events, we use a 3-day pairing window to capture peak NTR and RF, following similar studies (Couasnon et al., 2020; Kim et al., 2023). We also manually checked the RF and NTR time series of POT events and found that a 3-day window was generally sufficient to capture both peaks. To ensure event independence within the POT samples, previous studies have applied various declustering windows (e.g., 3 days (Haigh et al., 2016), 7 days (Santos et al., 2021), 10 days (Kim et al., 2023), and 14 days (Terlinden-Ruhl et al., 2025)). Longer declustering windows are often adopted when the influence of river discharge is present, as its effects can persist for several days (Terlinden-Ruhl et al., 2025). In this study, we use a 5-day declustering window (2.5 days before and after the event peaks), as elevated NTR rarely lasts more than 5 days at the Philadelphia tide gauge location. Previous studies have applied various search radii to identify TC events, such as ~400 km (Kim et al., 2023) and 500 km (Towey et al., 2022). In this study, we tested the sensitivity of the correlation between peak NTR and peak accumulated rainfall to the TC search radius, following Kim et al. (2023). Increasing the search radius captures more nearby TC tracks but also introduces events that are too distant to strongly influence flood drivers at the study site, thereby reducing the overall correlation between rainfall and NTR of the TC sample. We selected a 350 km search radius, as it leads to a higher correlation between flood drivers while still retaining a reasonable number of TC events in the sample."

Additional Refs:

Towey, K. L., Booth, J. F., Rodriguez Enriquez, A., and Wahl, T.: Tropical cyclone storm surge probabilities for the east coast of the United States: a cyclone-based perspective, Nat. Hazards Earth Syst. Sci., 22, 1287–1300, https://doi.org/10.5194/nhess-22-1287-2022, 2022.

Couasnon, A., Eilander, D., Muis, S., Veldkamp, T. I. E., Haigh, I. D., Wahl, T., Winsemius, H. C., and Ward, P. J.: Measuring compound flood potential from river discharge and storm surge extremes at the global scale, Nat. Hazards Earth Syst. Sci., 20, 489–504, https://doi.org/10.5194/nhess-20-489-2020, 2020.

Terlinden-Ruhl, L., Couasnon, A., Eilander, D., Hendrickx, G. G., Mares-Nasarre, P., and Antolínez, J. A. Á.: Accelerating compound flood risk assessments through active learning: A case study of Charleston County (USA), Nat. Hazards Earth Syst. Sci., 25, 1353–1375, https://doi.org/10.5194/nhess-25-1353-2025, 2025.

Page 8, Line 217-228: Could the authors clarify the physical justification for uniformly scaling entire NTR and RF time series based solely on peak values? For example, does this approach preserve key timing or intensity ratios in cases with asymmetrical hydrographs or localized RF bursts?

We thank the reviewer for raising this important point. In flood hazard modeling, the approach of scaling the observed flood driver time series to match design peaks and then considering them as design hydrographs/hyetographs with known return periods is already widely used. Examples include applications for NTR (Kim et al., 2023; Amorim et al., 2025), RF (Li et al., 2020; Kim et al., 2023), and streamflow (Yue et al., 2002). The assumption is that uniform scaling (also referred to as "same frequency amplification" in some studies) of the flood-driver time series produces another realistic time series.

Even with asymmetrical NTR time series/hyetograph shapes, our linear scaling approach preserves the relative timing and intensity ratios of sub-peaks to the main peak, which is a key factor in determining the resultant flooding. Linearly scaling a storm-tide hydrograph where tidal influence is more visible would create an unrealistic total water level time series. However, here we utilize NTR time series for scaling and then add tides and MSL to the scaled NTR time series in a consistent manner with their seasonal variability accounted for. Although we cannot validate synthetic events on a one-to-one basis against real events, we compared statistical properties of key time series characteristics, such as durations, peaks, intensities, and lag times, between observations and synthetic events (Fig. 9). These comparisons show that the synthetic events preserve the dependencies found in the observed events.

We acknowledge that scaling smaller events to much higher peaks can still produce unrealistic storm events. To mitigate this, we select observed events whose peak values are close to the target peak, so that the scaling factor remains close to 1 in most cases (with medians, for NTR: 1, for RF: 0.99), and we impose an upper limit on the scaling factor when generating events.

To clarify this further, and based on another reviewer's comments, we will revise the discussion section as follows:

L 493: "One key assumption of the framework is that uniform scaling (also referred to as "same frequency amplification") of flood-driver time series creates a realistic compound event. This approach has been widely adopted in previous studies to construct design hydrographs and hyetographs (e.g., Serafin et al., 2019; Moftakhari et al., 2019; Zellou and Rahali, 2019; Kim et al., 2023; Liu et al., 2024; Xu et al., 2024). However, assessing whether each generated synthetic storm event is physically plausible is not possible. Instead, we validated the framework by comparing statistical properties of key time series characteristics between observations and synthetic events (Fig. 9). Another  limitation of the proposed framework is that certain characteristics of synthetic events, such as RF duration and lag times, are limited to the observed values. To generate more diverse lag times, the observed lag times could be fitted to a parametric distribution (or alternatively to a copula that accounts for the dependence between peak values and lag times) and sample lag times from the fitted distribution during the event generation process. This would introduce unobserved lag times into the synthetic events, enhancing their diversity. Additionally, the stratification of POT events utilizes a simple yet commonly applied approach (e.g., Kim et al., 2023; Maduwantha et al., 2024) as discussed in Section 4.1. However, this method may fail to capture all TCs, particularly those that produce significant RF and storm surges from distances greater than 350 km. Such events are classified as non-TC events, meaning the analysis in Section 4.2 may not fully reflect the true characteristics of TC and non-TC events. "

Page 10, Line 273-275: The use of SFINCS is well-suited for handling large scenario sets; however, two model limitations warrant further discussion. First, SFINCS does not explicitly model nonlinear tide–surge interactions, which can influence the timing and amplitude of water levels in estuarine environments (e.g., Arns et al., 2020; Dullaart et al., 2023). Second, the omission of advection in the local inertia formulation may affect surge dynamics in narrow tidal channels like those surrounding Gloucester City. I recommend the authors provide a brief sensitivity analysis or comparison illustrating the impact of including vs. excluding the advection term, as SFINCS offers both options (Leijnse et al., 2021).

We agree with the reviewer that the tide-surge interaction can produce important effects on estuarine water levels. However, the limitations of SFINCS simulating these effects do not affect our analyses since the boundary condition of our model is placed along the Delaware, near the coastline of our study site. Therefore, we are not simulating the generation and propagation of the surge into the estuary, in which interaction between tides and surges occurs. Rather, we account for the tide-surge interaction within the presented framework to generate the boundary conditions for the flood model. The tide-surge interaction is accounted for by considering the dependency between NTR peaks with tidal high water peaks (i.e., the time difference between the NTR peak and the peak of the tide). As mentioned in L: 235 in the original manuscript, we use the observed time difference between peak NTR and the subsequent high tide of the sampled NTR time series to combine it with the sampled tidal signal. Therefore, the synthetic events reflect the effect of tide surge interaction, which was observed in the historical events. Fig. 1 below shows a histogram of the time difference between the peak NTR and the subsequent high tide of observed events (orange) and synthetic (blue) events.

[Figure]

Fig.1 The distribution of time difference between the peak NTR and the next high tide of observations (orange) and simulations (blue). Positive values indicate the peak NTR occurred before the next high tide.

As requested by the reviewer, to assess the potential impact of omitting the advection term in the flood model runs, we conducted an additional set of simulations with the advection term enabled. The results (see Fig.2 below) show negligible differences in resultant flooding compared to our original simulations without advection, even in the narrow tidal channels surrounding Gloucester City. Given these results and the fact that the main conclusions of our study are based on the statistical framework of boundary condition generation rather than fine-scale hydrodynamic sensitivities, we are confident that the omission of the advection term does not largely affect the main results and conclusions of the paper.

To further clarify this and address a related concern raised by another reviewer, we will add the following sentences.

L 465: "At the Philadelphia tide gauge, peak NTR often occurs 4–7 hours before the next high tide (see Fig.S2 in the supplementary material). To account for this, we combine scaled NTR time series with tide time series using the observed lag between peak NTR and subsequent high tide of the sampled NTR event (see Section 4.3.3)."

We will revise the following sentence.

L 274: The model is run with the advection term neglected, solving the local inertia equations (we tested the sensitivity of the results when the advection term was enabled, but changes were negligible).

[Figure]

Fig 2. Changes in flood depths associated with variability when the advection term is enabled, in MSL (left) and Tide (right) of a selected 0.01 AEP most-likely event. (a) When considering the lowest 30-day MSL, (b) when considering the tidal segment with the lowest high tide, (c) when considering the highest 30-day MSL, (d) when considering the tidal segment with the highest high tide, (e) the difference between (a) and (c), (f) difference between (b) and (d). (X, Y coordinates system: UTM- zone 18N).

Page 11, Line 300-301: The authors use a fixed NTR threshold of 0.63 m to yield ~5 exceedances per year, which is reasonable and aligns with past compound flood studies. However, the threshold selection could be strengthened by applying one of several recent automated, data-driven approaches developed for POT analysis, such as the Sequential Goodness-of-Fit method (Bader et al., 2018), the Extrapolated-Height Stability method (Liang et al., 2019), the L-moment Ratio Stability method (Silva Lomba & Fraga Alves, 2020), or the comparative multi-method approach applied in a coastal flood design context by Radfar et al. (2022).

We thank the reviewer for the comment and agree that more advanced approaches, such as those mentioned, offer more robust methods for objectively determining thresholds. However, these methods are primarily designed to optimize the fit of the distribution tail. In the context of compound flooding, extreme flood events are not always driven by extreme peak flood driver combinations. With a suitable combination of timing, duration, intensities, and tidal conditions, extreme flooding can also occur under moderate peak NTR and RF combinations (Santamaria et al., 2025). To generate synthetic storm events with moderate NTR and RF peaks, the copulas must therefore be fitted using POT events with flood potential and still large enough to be well described by a GPD (Generalized Pareto Distribution). Accordingly, we selected a threshold that balances both, small enough to capture moderate conditions, yet large enough to represent the upper tail of the distribution. Rather than applying more robust threshold selection methods, we followed the approach used in similar studies and considered the average number of historical flood events to guide our choice.

We will add the following text to the manuscript.

L 161: Recent data-driven threshold-selection methods such as the Sequential Goodness-of-Fit method (Bader et al., 2018), the Extrapolated-Height Stability method (Liang et al., 2019), L-moment ratio stability (Silva Lomba & Fraga Alves, 2020), and a comparative multi-method approach (Radfar et al., 2022) provide robust POT thresholds but primarily optimize tail fit. Extreme compound flood events are not necessarily generated by extreme flood driver peaks. With favorable timing, duration, and tidal conditions, extreme flooding can occur even under moderate flood-driver conditions (Santamaria et al., 2025). Therefore, we set thresholds on NTR and RF, targeting an average of five exceedances per year to extract POT events, following similar studies (Jane et al., 2020; Kim et al., 2023).

Page 12, Line 328-336: While the authors maintain stratification for joint probability estimation, they combine TC and non-TC time series for event generation based on overlapping confidence intervals and similar time series shapes. Given the well-established physical differences between tropical and extratropical systems (precipitation structures, spatial scales, storm tracks), could the authors clarify how confident they are that this approach adequately preserves the distinct characteristics of these storm types?

We agree that it is well established that tropical cyclones, extratropical cyclones, and other non-cyclonic locally generated systems differ in terms of physical properties. However, for our study area, factors such as the limited sample size of TC events, the small size of the catchment, and the hydrologic response of

the upstream Delaware River reduce the extent to which the inherent large-scale storm characteristics are reflected in the time series of NTR and basin-average rainfall.

The overlapping confidence intervals of Kendall's tau, along with comparisons of distribution parameters fitted to the time series characteristics of RF and NTR, indicate no statistically significant differences between the TC and non-TC samples (see Fig. 6). Similarly, the shapes of the NTR and basin-average RF time series produced by TCs are similar to those generated by non-TC events (see Fig. 7). Based on these results, we conclude that generating event time series (i.e., water-level hydrographs and RF hyetographs) separately for the two storm types would yield results similar to those obtained without differentiating between them. However, we acknowledge that in other locations, significant differences in time series characteristics between TC and non-TC samples may exist, especially when model domains are large. In those cases, we recommend conducting event generation separately for each storm type. This limitation is already discussed in the original manuscript (Section 6, Lines 440–455).

Additionally, given the limited number of TC events, how do the authors assess whether their analysis has sufficient statistical power to detect meaningful differences? Would alternative approaches like physics-based conditioning (e.g., storm track or seasonal constraints) potentially better preserve known meteorological distinctions while addressing sample size limitations?

We thank the reviewer for this thoughtful comment. We acknowledge that the limited number of TC events in our dataset reduces the statistical power to detect subtle differences in time series characteristics between TC and non-TC samples. As noted in the manuscript (Section 6, Lines 440–455), this is recognized as a limitation of our application.

Methods involving physics-based models to generate synthetic flood drivers from TC events (often using synthetic TC tracks) can better preserve the inherent spatio-temporal characteristics of storms (e.g., Emanuel et al., 2006; Gori et al, 2020). However, these methods are computationally demanding, as they require both hydrologic and hydrodynamic modeling, which limits their ability to generate large numbers of events. We agree that such approaches could help address the limited TC sample size in our study. Implementing this type of framework is beyond the scope of the current work, but we plan to incorporate it in future studies.

We plan to add the following sentences to the main manuscript.

L 452: "One option to overcome the limited TC sample size in our analysis is employing physics-based models to generate time series of flood drivers from synthetic TC tracks (e.g., Emanuel et al., 2006; Gori et al., 2020). We plan to explore this in future work."

Page 20, Figure 10: The authors demonstrate substantial flood depth due to MSL and tidal variability using a single most-likely 0.01 AEP event. While this effectively illustrates the potential importance of these factors, could the authors comment on whether this sensitivity pattern is representative across different event types and return periods?

The most-likely 0.01 AEP (100-year return period) event was selected to show an illustrative example to highlight the potential influence of MSL and tidal variability on resultant flooding. While traditional approaches often neglect this variability in their modeling frameworks, we explicitly account for it. We use the 0.01 AEP, as it is commonly adopted in flood hazard mapping for planning purposes (e.g., FEMA Special Flood Hazard Areas; FEMA, 2020).

We agree that this single event may not fully represent the sensitivity patterns across the full range of event types and return periods. In a separate study (Santamaria et al., 2025, under review), we simulated flooding for the same study site using a set of 5,000 synthetic storm events that were generated from the developed statistical framework. Resultant flooding shows a wide range of variability, even among events with similar joint return periods. However, attributing this variability to a single factor like MSL or tides is challenging due to the complexity of their interactions. This is discussed in detail in Santamaria et al. (2025).

For discussing this in the manuscript, we will modify the section starting from L 488 as follows:

L488: "Analyzing only the most likely event, even if it appears to be the most plausible based on observations, does not capture the range of flood levels that could be generated by different combinations of flood drivers (i.e., NTR and RF) with different time series properties. Therefore, the flood model simulations presented here are aimed at evaluating the importance of explicitly accounting for the variability of MSL and tides, and not to produce comprehensive probabilistic flood maps. In a separate study (Santamaria et al., 2025), we simulated flooding of 5,000 synthetic storms at this site and found large variation in resultant flooding, even for events with similar joint return periods. However, attributing this variability to a single factor like MSL or tides is challenging due to the complexity of their interactions."

Additional Ref.

Federal Emergency Management Agency (FEMA). (2020). Flood Insurance Study Guidelines: Guidelines and Specifications for Flood Hazard Mapping Partners. https://www.fema.gov/flood-maps/guidance-partners/guidelines-specifications

Santamaria-Aguilar, S., Maduwantha, P., Enriquez, A. R., and Wahl, T.: Large discrepancies between event- and response-based compound flood hazard estimates, EGUsphere [preprint], https://doi.org/10.5194/egusphere-2025-1938, 2025.

Additionally, given that the most pronounced effects occur along the Delaware River and Newton Creek boundaries, could the authors discuss whether the model's spatial resolution, boundary condition placement, or coastal setup might be influencing the magnitude of these sensitivities?

In the flood model, the coastal boundary is placed along the Delaware River to match the water levels at the Philadelphia tide gauge and includes the entire catchment of Newton Creek to capture all contributing runoff. We used the Coastal National Elevation Database (CoNED) digital elevation model with a horizontal resolution of 1 meter and a vertical accuracy of 10 cm. We use the subgrid approach of SFINCS with a dual resolution of 10m and 1m. This level of detail is relatively high compared to many other compound flood hazard assessments, which often use coarser grids. Given the size of the Delaware River and surrounding creeks, we believe the model adequately resolves hydrodynamics and that the observed sensitivities reflect physical processes rather than numerical limitations.

Refs.:

Arns, A., Wahl, T., Wolff, C., Vafeidis, A. T., Haigh, I. D., Woodworth, P., Niehüser, S., & Jensen, J. (2020). Non-linear interaction modulates global extreme sea levels, coastal flood exposure, and impacts. Nature Communications, 11, 1918.

Dullaart, J. C. M., Muis, S., de Moel, H., Ward, P. J., Eilander, D., & Aerts, J. C. J. H. (2023). Enabling dynamic modelling of coastal flooding by defining storm tide hydrographs. Natural Hazards and Earth System Sciences, 23, 1847–1862.

Leijnse, T., Dazzi, S., Yu, D., & Bates, P. D. (2021). Efficient coastal flood hazard mapping with a 2D non-inertia model. Coastal Engineering, 170, 103994.

Bader, B., Yan, J., & Zhang, X. (2018). Automated threshold selection in extreme value analysis via goodness-of-fit tests with adjustment for false discovery rate. Annals of Applied Statistics, 12(1), 310–329.

Liang, B., Shao, Z., Li, H., Shao, M., Lee, D., 2019. An automated threshold selection method based on the characteristic of extrapolated significant wave heights. Coast. Eng. 144, 22–32.

Radfar, S., Shafieefar, M., & Akbari, H. (2022). Impact of copula model selection on reliability-based design optimization of a rubble mound breakwater. Ocean Engineering, 260, 112023.

Silva Lomba, J., Fraga Alves, M.I., 2020. L-moments for automatic threshold selection in extreme value analysis. Stoch. Environ. Res. Risk Assess. 34 (3), 465–491.

Gao, S., Zhang, J., Li, D., Jiang, H., & Fang, Z. N. (2021). Evaluation of multiradar multisensor and stage IV quantitative precipitation estimates during Hurricane Harvey. Natural Hazards Review, 22(1), 04020057.

Gomez, F. J., Jafarzadegan, K., Moftakhari, H., & Moradkhani, H. (2024). Probabilistic flood inundation mapping through copula Bayesian multi-modeling of precipitation products. Natural Hazards and Earth System Sciences, 24(8), 2647-2665.

---

## Author Comment (AC2)

The manuscript "Generating Boundary Conditions for Compound Flood Modeling in a Probabilistic Framework" by Maduwantha et al. introduces a statistical framework designed to generate many synthetic but physically plausible compound events, including storm-tide hydrographs and rainfall fields, which can serve as boundary conditions for dynamic compound flood models. The framework is later applied to the case of Gloucester City in New Jersey.

The topic of the manuscript is highly relevant for quantifying flood hazard, particularly water depth resulting from the joint occurrence of storm surge and rainfall. However, the proposed framework requires further clarification and additional information to assess its validity better and ensure reproducibility by others.

We thank the reviewer for their constructive and insightful comments, which have helped enhance the overall quality of our manuscript. Below, we provide detailed responses to each point and outline how we plan to address them in the revised manuscript. The changes to the existing text are highlighted using track changes with line numbers in the original manuscript.

The novelty of the proposed framework should be better highlighted. If I understand correctly, the essence of the proposed framework is to select joint events of NTR and rainfall from historical observations, and then fit a bivariate copula to generate "unseen" pairs. Pairs are used to amplify historical time series of NTR and rainfall over a short period of time with a temporal resolution consistent with the one required by the hydrodynamic model. An almost identical workflow was proposed by Xu, H et al (2024) "Combining statistical and hydrodynamic models to assess compound flood hazards from rainfall and storm surge: a case study of Shanghai", Hydrol. Earth Syst. Sci., 28, 3919–3930, https://doi.org/10.5194/hess-28-3919-2024. How does this framework differ from Xu et al 2024? How does this framework differ from previous studies? The novelty is hidden in the introduction.

We thank the reviewer for raising this important point. While our framework shares some conceptual similarities with Xu et al. (2024), it introduces several methodological and practical novelties that distinguish it from previous studies.

We agree that the general workflow of (i) modeling the joint probability distribution of flood drivers, (ii) sampling unseen pairs from the fitted distribution, (iii) assigning corresponding time series, and (iv) propagating them through a compound flood model is not unique to our study. This general approach has been adopted in several previous works (e.g., Serafin et al., 2019; Moftakhari et al., 2019; Zellou and Rahali, 2019; Kim et al., 2023; Liu et al., 2024; Xu et al., 2024), and we do not claim novelty in this regard. These studies, including Xu et al. (2024), were primarily tailored to generate design events with specified joint return periods (e.g., 100-yr,50-yr) of flood-driver peaks. This only supports an "event-based" hazard analysis, where one or a few design events are simulated through flood models, and the joint probability of the flood drivers is assumed to be the same as the probability of the flood response.

In contrast, our framework is designed to generate a large and diverse set of synthetic storm events. These are then propagated through the hydrodynamic model to produce a broad ensemble of flood responses. Extreme value analysis is then applied directly to the resultant flood depths, thereby supporting a response-based flood hazard assessment. This shift from generating boundary conditions for event-based

to response-based analysis is a key novelty of our work, as it allows us to capture the full spectrum of possible flood outcomes, rather than relying solely on the flood response of a single design event or a small subset of events. Additionally, our framework can also be used to support event-based hazard modelling by generating many diverse synthetic storm events with a unique (or known) joint return period. In addition to the objective differences, we have listed other key advancements of our modeling framework over Xu et al. (2024):

1. Accounting for distinct storm types (populations), thereby capturing the unique dependence structures of flood-driver peaks
2. Generation of total water level time series by combining scaled NTR, tides, and MSL consistent with seasonal variations
3. Offers a more flexible framework, by incorporating different rainfall accumulation times and employing a two-way conditional sampling method that enables modeling of a wide range of storm events, including non-extremes

To better highlight the novelty of our work, we have revised the following paragraphs in our introduction.

L 75: Rescaling of total water level (or non-tidal residuals (NTR) time series) of observed events is another deterministic approach that leverages observed event data (Dawson et al., 2005; Kim et al., 2023; Xu et al., 2024).

L 100: "Among the applications of uniform scaling of flooding drivers, Xu et al. (2024) applied the "same frequency amplification" method to construct a 200-yr storm surge hydrograph and rainfall hyetograph for their flood simulations. However, their approach was limited to point rainfall and assumed uniformly distributed rainfall across the catchment. Kim et al. (2023) proposed a framework for generating synthetic time series of RF fields and associated NTR by scaling time series of observed TC events. The framework was used to capture different spatial patterns of RF fields as this aspect was shown to significantly contribute to compound flood hazard (e.g., Gori et al., 2020). However, their analysis exclusively focused on TC events and the methodology only produces NTR time series and does not extend to producing complete storm-tide hydrographs; this is because it was applied to the Texas coast where the tidal range is small, and where compound flooding is primarily driven by TCs. Other types of storms can produce compound flooding in many other areas and tides often contribute significantly to the resulting still water levels.

The existing statistical approaches that generate time-varying boundary conditions for dynamic compound flood models are primarily intended to construct design events with specified joint return periods (e.g., 50-yr, 100-yr) (Serafin et al., 2019; Moftakhari et al., 2019; Zellou and Rahali, 2019; Kim et al., 2023; Liu et al., 2024; Xu et al., 2024). This method supports the "event-based" flood hazard analysis, where a single (or only a few) synthetic event with known joint return periods is simulated through a flood model, and it is assumed that the (joint) probability of the flood drivers directly translates into the probability of the flood response. However, this neglects the range of potential different flooding scenarios that may arise from variations in temporal and spatial patterns, differences in the relative timing of multiple flood drivers, and other complex interactions (for example, tide-surge interactions). Additionally, their approach is

. For a more complete characterization of flood hazard and risk, the flood response of many synthetic events needs to be modeled, allowing the derivation, for example, of return levels of flood depth at all points within the model domain (i.e., "response-based" flood hazard analysis)."

In lines 502-503, the Authors say that "Although measures are taken to prevent the generation of physically unrealistic events (see Section 4.3), it cannot be fully ruled out." If this statement is true, then the Authors cannot claim that the framework generated physically plausible compound events (Abstract - Lines 16-17). This is quite an important point, and the Authors need to be transparent about the potential of the framework to generate physically plausible events or not.

We thank the Reviewer for the comment. Our claim in the Abstract (Lines 16–17) that the framework generates physically plausible compound events comes from a statistical and process-based perspective since the framework explicitly preserves the dependence structures of peaks of flooding drivers and adheres to ranges and interdependencies of observed timeseries characteristics, as described in Section 4.3.

However, we acknowledge that we cannot fully eliminate the risk of generating extreme storm events that may be physically "implausible", at least under current climate conditions (we believe this is not unique to our approach but true for most statistical methods that generate extreme unseen environmental data). For instance, when deriving peak NTR–RF combinations from the multivariate statistical model, the use of unbounded marginal distributions can yield large peaks that might be physically impossible to occur. However, the absence of such events in the observational record does not in itself indicate that the generated synthetic events are unrealistic, as the record length may be insufficient to capture their occurrence. Extremes are often wrongly perceived as impossible until they happen. Moreover, even with the assumption that uniform scaling produces realistic NTR and RF time series individually, it still requires the additional assumption that their combined representation is also physically realistic. Ideally, a direct one-to-one validation of synthetic events against observed events would provide a rigorous test of these assumptions, but such validation is impossible. To evaluate the performance of our framework, we compared the statistical properties of key time-series characteristics, including durations, peaks, intensities, and lag times between observations and synthetic events (Fig. 9).

To further acknowledge these limitations, along with comments from another reviewer, we will add the following to the discussion:

L 494: "One key assumption of the framework is that uniform scaling (also referred to as "same frequency amplification") of flooding-driver time series creates a realistic compound event. This approach has been widely adopted in previous studies to construct design hydrographs and hyetographs (e.g., Serafin et al., 2019; Moftakhari et al., 2019; Zellou and Rahali, 2019; Kim et al., 2023; Liu et al., 2024; Xu et al., 2024).

However, assessing whether each generated synthetic storm event is physically plausible is not possible. Instead, we validate the framework by comparing statistical properties of key time series characteristics between observations and synthetic events (Fig. 9).

The data selection procedure and its effects on the results need further clarification.

We appreciate the Reviewer's comment and agree that the data selection procedure requires a clearer explanation. The framework is designed to generate high-resolution rainfall fields over the catchment and storm-tide hydrographs at the coastal boundary. For water levels, we use the nearest tide gauges to the study site (Philadelphia (St. ID: 8545240) and Philadelphia Pier 11-north (St. ID: 8545530)). Although some rainfall gauges exist near Gloucester City, the Philadelphia Airport rain gauge provides the longest hourly record (from 1900). For rainfall fields, radar-based products such as MRMS offer higher resolution and accuracy compared to other gridded RF data (1 km, hourly), but their temporal coverage is limited (MRMS: from 2012; Stage IV: from 2002). We therefore use AORC precipitation data (1979 to present), which balances high resolution (4 km, hourly) with sufficient temporal coverage (from 1979) to extract observed RF fields from many events. For identifying TC-induced POT (peak over threshold) events, we use HURDAT2, the most widely used and reliable TC catalog for the Atlantic basin.

While using different datasets may introduce some changes to the joint probability distributions and resulting synthetic storm events, we emphasize that our choices represent the best available balance of resolution, accuracy, and temporal coverage. Therefore, we believe our results are robust. A full sensitivity analysis of synthetic storms to different datasets would be valuable, but it's beyond the scope of this study.

We will revise the relevant sections in the original manuscript as follows:

L 135: "For the statistical analysis, we consider RF and NTR as flood drivers. We use hourly water level data from  the  National Oceanic and Atmospheric Administration (NOAA) Philadelphia (St. ID: 8545240) and Philadelphia Pier 11-north (St. ID: 8545530).

L 144: "We use both gridded RF data from the Analysis of Period of Record for Calibration (AORC) from 1979 to 2021 and hourly RF gauge data at the Philadelphia International Airport from 1900 to 2021 (Kitzmiller et al., 2018). Although radar-based quantitative precipitation estimates, such as the Multi-Radar Multi-Sensor (MRMS) products, often provide higher accuracy compared to other gridded rainfall products, their temporal coverage is relatively short (Gao et al., 2021; Gomez et al., 2024). We use AORC rainfall data because of its availability from 1979 and its demonstrated higher accuracy among products with similar temporal coverage, while offering hourly data with ~4 km spatial resolution (e.g., Hong et al., 2024; Kim and Villarini, 2022) ~~AORC RF data has demonstrated higher accuracy compared to other gridded data sets while offering an hourly temporal resolution and ~4 km spatial resolution (e.g., Hong et al., 2024; Kim and Villarini, 2022)~~."

First, synchronous NTR are selected, and later on astronomical tide and mean sea level are added. Did the Authors consider using the concept of skew surge? If not, why? In addition, tide-surge interaction is mentioned but not really discussed. How relevant is it? Would the concept of skew surge solve it?

We thank the Reviewer for the comment. We agree that skew surge would implicitly account for tide–surge interactions and offers a straightforward way to combine it with Tides. However, as skew surge provides only two values per tidal cycle (semidiurnal in the Philadelphia region), it cannot capture the full temporal evolution of NTR at hourly scales or the timing between flood drivers, which are critical for compound flooding (Gori et al., 2021). Previous studies (e.g., Terlinden-Ruhl et al., 2025) assumed a constant skew surge over a 12-hr period, which we consider insufficient to reproduce realistic hydrographs for our analysis. Additionally, estimating the duration of elevated NTR is difficult from skew surge alone, but it is a key time series characteristic that we want to examine and make sure it is consistent in the synthetic events.

At the Philadelphia tide gauge, peak NTR often occurs 4–7 hours before the next high tide (see Fig.1 below). To account for this, we combined scaled NTR time series with tide time series using the observed lag between peak NTR and subsequent high tide of the sampled NTR event (see Section 4.3.3), ensuring that synthetic events preserve the same dynamics.

[Figure]

Fig.1 The distribution of time difference between the peak NTR and the next high tide of observations (orange) and simulations (blue). Positive values indicate the peak NTR occurred before the next high tide.

In addition to the explanation in the main manuscript in Section 4.3.3, we plan to further clarify this in the discussion by adding the following:

L 465: "At the Philadelphia tide gauge, peak NTR often occurs 4–7 hours before the next high tide (see Fig.S2 in the supplementary material). To account for this, we combine scaled NTR time series with tide time series using the observed lag between peak NTR and subsequent high tide of the sampled NTR event (see Section 4.3.3)."

Second, multiple rainfall measurements are considered. However, it is unclear how such measurements are aggregated and how this aggregation affects the dependence between NTR and rainfall, and so the fitted copula. Moreover, which rainfall measurement is used as a reference for the lag time between peak surge and peak rainfall?

A detailed description of the rainfall (RF) data treatment is provided in our previously published work (Maduwantha et al., 2024), but we summarize the main steps in the Methods section. Rain gauges measure very local weather conditions. However, the assumption that such point RF quantities are uniformly distributed over the entire catchment could lead to mischaracterization of the flood hazard potential. Therefore, we apply a bias correction to the hourly RF gauge data to match the hourly basin-averaged RF quantities calculated from AORC. The quantile mapping method is used for the bias correction, fitting both hourly measured gauge data and hourly AORC basin-averaged data to gamma distributions. This procedure allows us to increase the POT sample size, thereby reducing uncertainty in the marginal distributions. As discussed in Maduwantha et al. (2024), the correlation between peak NTR and rainfall shows nonstationarity with an increase in recent decades. Hence, we estimated the copula parameters using POT events from the last 30 years, ensuring that the dependence structure reflects current climate conditions, and we avoid potential underestimation of the flood hazard.

Lag times and other time-series characteristics (e.g., durations, peaks, and intensities) are derived directly from the historical events, using AORC rainfall fields combined with tide gauge records (those are then scaled to match the target peaks sampled from the NTR-RF joint distribution). This part is explained in the original manuscript from L 185 to L 190.

To further clarify how the aggregation of RF gauge data and AORC data affects dependence, we will revise the following paragraph:

L 147: "Rain gauges measure highly localized rainfall. Assuming that these point measurements occurred uniformly distributed across the entire catchment can misrepresent the compound flood hazard. To address this, we apply a bias correction to the hourly gauge data so it matches the basin-averaged hourly rainfall estimates derived from AORC. This correction is performed using the quantile mapping method, in which both the gauge-based and AORC-based rainfall distributions are fitted to gamma functions. To leverage the long-term in-situ observations and obtain more robust results from the statistical analysis, we apply a bias correction to the hourly RF gauge data, to match with the hourly basin-average RF values calculated from AORC. The bias correction is performed using the quantile mapping method, fitting both the hourly measured gauge data and the hourly AORC basin-average data to gamma distributions (for more details see Maduwantha et al. (2024))."

We plan to add the following:

L 171: "Maduwantha et al. (2024) found significant non-stationarity in Kendall's $\tau$ between peak NTR and RF over the analysis period. To capture most recent climate conditions and avoid underestimating compounding effects, we model dependence using only the last 30 years of data."

Finally, the distinction between TC and non-TC leads to copulas with different asymmetries. How do the Authors justify such differences? What happens when the TC and non-TC are combined together? I would say this is mostly relevant in paragraph 5.3 "Event Generation Process". How do the Authors know that the 106-year event (which is also quite an interesting number!) corresponds to a TC? Given the length of the data, the 106-year event is inferred from the copula and not observed. Regardless of how the Authors track whether it is a TC or a non-TC, how sensitive are the results to the type of event? For example, what is the difference between a TC and non-TC event with the same return period? What about in terms of the drivers' magnitude and water depth? I suggest adding some sensitivity analysis to the assumptions made, including the lag time between peak rainfall and peak surge.

We thank the Reviewer for the comment. A key objective of our framework is to account for the distinct dependence structures of different storm types (TCs vs. non-TCs), which are often overlooked when all POT events are treated as a single population. In our analysis, peak NTR and peak RF were found to be more strongly correlated in TC-induced events than in non-TC events, in line with previous studies (e.g., Kim et al., 2023). Therefore, as the reviewer mentioned, TC and non-TC lead to copulas with different asymmetries. We derive 5,000 combinations of peak NTR and RF by sampling from the fitted copulas to each sample such that the relative proportion of extremes is consistent with their historical occurrence. 351 combinations from the copulas fitted to the TC sample and 4,649 from the fitted copulas to the non-TC sample.

The illustrative "106-year event" presented in the paper was drawn from the TC sample (one of the 351 combinations). Since the events are generated from the fitted copulas through the Monte Carlo approach, the resulting combinations do not correspond to standard return periods (e.g., exactly 50- or 100-year), instead, reflect the joint probability distribution of peak NTR and RF.

Based on the analysis we conducted in section 4.2, we found no significant differences in statistical features nor various time series characteristics in NTR and RF from TC events and non-TC events. Given these results, we conclude that generating the event time series (i.e., water level hydrographs and RF hyetographs) separately for the two storm types would produce similar results compared to the ones we derive without stratifying. Yet, we acknowledge that with sufficient data, the most effective approach would be to use observed time series of flood drivers from TC events exclusively for generating synthetic TC events, while using those from non-TC events separately to generate synthetic non-TC events.

The NTR and RF peak magnitudes of an event with the same joint probability can differ substantially between storm types and depending on where it lies along the probability isoline. For example, the selected event (NTR = 1.75 m; 18-h RF = 80 mm) has a ~106-yr joint return period when joint probability distributions of TC and non-TC storms are combined. The same event has a ~111-yr joint return period in the TC sample and ~2431-yr joint return period in the non-TC sample. Conversely, an event with relatively lower peaks (NTR = 1.0 m; 18-h RF = 50 mm) has a ~5-yr joint return period in the non-TC sample but ~15-yr joint return period in the TC sample. In conclusion, small to moderate events are more frequent in the non-TC storms, whereas the most extreme events are more likely generated by TCs. This aspect is discussed in detail in Maduwantha et al. (2024).

As discussed in L 464, lag times between peak NTR and peak RF can vary considerably, and more extreme events tend to exhibit shorter lag times (see Figs. 4 (e) and 4 (i)). However, this behavior is evident in both TC and non-TC events. To reflect this behavior in synthetic events, we select nearby historical events for scaling and adopt the associated lag time from one of the selected time series. This ensures that synthetic

compound events retain the same timing dynamics as similar observed events (See Fig. 9). Extending this analysis to evaluate how differences between storm types influence flood depths would require flood model simulations of a broader set of events, which is beyond the scope of this study.

For further clarification, the following sentences will be revised:

L 239: "Target events are derived from copulas fitted to the TC sample and the non-TC sample. If the target event is derived from a copula fitted to TC (non-TC) events, we sample the month from the distribution of TC (non-TC) events (Fig. 2 (a)). Once the month is selected, we randomly sample a MSL value and a tidal signal segment from the selected month (Fig. 2 (g))."

The following will be added:

L 420: "The joint probability distributions of peak NTR and peak RF of TC and non-TC storms are substantially different. Small to moderate compound events are more frequent in the non-TC storms, whereas the most extreme compound events are more likely generated by TCs. This aspect is further discussed in Maduwantha et al. (2024).

Minor comments.

The "target event" is never explicitly defined, and from my personal perspective, this creates some confusion. How is it how is it selected?

We thank the reviewer for raising this point. We refer to a "target event" as a synthetic peak NTR–RF combination selected from the 5,000 combinations generated via Monte Carlo sampling of the fitted copulas. While some studies use the term "design event", we used "target event" since our framework does not rely on standard design return periods (e.g., 50- or 100-year). We will define it in L 179.

L 179: "We generate an event set of 5,000 combinations of NTR and RF ("target events") by sampling from the fitted copulas such that the relative proportion of extremes is consistent with the empirical distribution."

ETC and non-TCI seem to be used interchangeably. I suggest checking the notation for consistency.

Checked and will be corrected.

Line 409: the Authors say that flood depth varies in some regions. However, the case study seems to concern only one region. I would suggest checking this sentence.

The term region refers to the different areas within the catchment. We will revise the sentence as follows:

"The difference in flood depths between using the highest and lowest 30-day MSL reaches up to 1 m in some areas of the city".

The comparison between rainfall and surge is done considering duration. How did the Authors handle discrete variables when assessing correlation?

We thank the reviewer for raising this point. We have used the MATLAB function "corr" for calculating Kendall's tau. When handling ties, this function uses an adjustment, calculating tau-b, which still varies in the range of -1 to +1 and leads to a stable estimation. Tau-b is calculated as follows:

$$Tau_b = \frac{C - D}{\sqrt{(C + D + T_x)(C + D + T_y)}}$$

Where,

$C$ = number of concordant pairs
$D$ = number of concordant pairs
$T_x$ = number of pairs tied only in X
$T_y$ = number of pairs tied only in Y

---

## Author Response (AR2)

**Response to the referee 1**

The authors have addressed all major concerns. The manuscript is significantly improved. I recommend acceptance pending only one minor revision:

1. The parenthetical addition about advection testing could reference the new Figure 2 in supplementary material: "(we tested the sensitivity of the results when the advection term was enabled and found negligible differences; see Fig. S5)" Currently, it is missed.

We sincerely thank the reviewer for noticing this. Although we have included that figure in our previous reply to the reviewer, it is missing in the submitted supplementary material. This has now been corrected, and the updated figure is included in the revised supplementary file.

**Response to the referee 2**

First of all, I would like to thank the reviewers for carefully addressing my comments. However, two points remain unclear.

1. First, about the process in itself. The Authors say that "we found no significant differences in statistical features nor various time series characteristics in NTR and RF from TC events and non-TC events. Given these results, we conclude that generating the event time series[…] separately for the two storm types would produce similar results compared to the ones we derive without stratifying". Yet, you mentioned 3 different copulas in the generative process, with opposite tail dependence in the case of TC and non-TC (from line 340). How do the Authors justify that the results with and without stratification are the same? What is then the added value of the methodology proposed? As a more general comment, it would help to use different colours for the simulated pairs in Figure 3 panel b, as done in panel a.

We sincerely thank the reviewer for these insightful questions and thoughtful feedback. We acknowledge that some confusion may remain regarding the characterization of storm events, and we appreciate the opportunity to clarify this aspect of the framework.

In this study, we consider two main categories of storm-event characteristics derived from historical observations:

(a) Peak values of non-tidal residuals (NTR) and rainfall (RF)
(b) Other time-series characteristics (e.g., temporal evolution, duration, lag time, intensity)

Accordingly, the proposed framework consists of two main steps. First, we derive the joint probability distribution of peak NTR and peak RF, and simulate peak combinations from it, which addresses category (a). Second, we assign a time series to the simulated peak combination, which addresses category (b).

We check how both categories of characteristics vary between the two storm types, tropical cyclones (TCs) and non-TC events. Notably, the dependence (here we quantified using Kendall's tau) between peak NTR and peak RF is substantially stronger in the TC sample than in the non-TC sample (see Figure R1 below). As a result, treating all events as a single sample when estimating joint probabilities would mischaracterize

the dependence structure. For this reason, the joint probability analysis is performed separately for TC and non-TC events. Within each storm type, we further consider two conditional samples (conditioning on NTR and conditioning on RF), and we identify and fit the copula family that best represents each conditional dependence structure. Figure R2 illustrates these conditional samples and the corresponding joint probability distribution for each conditional sample and combined for storm type. This part of the analysis has already been published in Maduwantha et al. (2024), so we do not plan to include these figures in the main manuscript.

When examining the other time-series characteristics of the two storm types, we find no statistically significant differences in lag times, durations, or intensities (see Figures 4–7 in the main manuscript). This finding supports our conclusion that, when selecting an observed historical time series for scaling, it is not critical whether the event originates from a TC or non-TC storm, as the temporal characteristics are comparable. Thus, we claim, "generating the event time series (i.e., water level hydrographs and RF hyetographs) separately for the two storm types would produce similar results compared to the ones we derive without stratifying". However, for estimating the joint probability distribution of peak values, stratification by storm type remains necessary due to the different dependence structures.

In other catchments, where significant differences exist in the time series characteristics between TC and non-TC samples (as discussed in Section 4.2 of the main manuscript), we suggest that the event generation process could also be conducted separately for each storm type using the developed framework.

We have revised the manuscript to improve clarity on this point, as follows. Additionally, we have updated Figure 3 (panel b) to show simulated pairs in separate colors for each storm type for better comparison, as suggested (see Figure R3).

L496: Given these results, we conclude that  sampling and scaling of the observed event time series (i.e., water level hydrographs and RF hyetographs) separately for the two storm types would produce similar results compared to the ones we derive without stratifying. Note that stratification is still applied when deriving the joint probability distribution since the dependence structure of the peaks of NTR and RF is substantially different. Importantly, this applies to the specific study location. In other places, significant differences may exist in the time series characteristics between TC and non-TC samples (as discussed in Section 4.2), warranting that the event generation process is also conducted separately for each storm type.

2. Second is about the 106-year event. The Authors say: "The selected event (NTR = 1.75 m; 18-h RF = 80 mm) has a ~106-yr joint return period when joint probability distributions of TC and non-TC storms are combined. The same event has a ~111-yr joint return period in the TC sample and ~2431-yr joint return period in the non-TC sample." Based on the sampling, most of the events come from the non-TC case (roughly 93%). I would have expected a return period from the combined sample closer to the non-TC case. This is also what the Authors say about the fact that a TC event is less frequent than a non-TC. However, the return period of the event from the combined set (106 years) is closer to the return period of the TC sample (110-yr). How do the Authors justify this result?

While it is true that approximately 93% of our events are related to non-TC sources, the return period of the selected event (~106 years in the combined distribution) is much closer to the return period estimated using only the TC sample (~111 years), rather than the non-TC sample (~2431 years). This result arises from the shape of the joint probability distributions of individual storm populations and how the combined joint probability distribution is constructed.

Although non-TC events are more frequent in the historical record, TC events are often more extreme and exhibit stronger dependence between peak NTR and RF. This behavior is reflected by the structure of the joint probability density, particularly in the upper tail. The selected event (NTR = 1.75 m; RF = 80 mm) lies in a region of the joint probability space where TC events have a higher contribution to the total exceedance probability (see Figures R2 and R3 where the selected event is shown as a star). As a result, the combined annual exceedance probability (AEP) at this point is more influenced by the TC distribution, leading to a return period that is numerically closer to the TC-only estimate.

This can also be explained using the equations that we used to derive the total joint probability distribution of two storm type populations. Assuming these two populations are independent from each other, the total annual non-exceedance probability ($ANEP$) of a given pair of $(NTR, RF)$ is calculated as follows (Maduwantha et al., 2024):

$$ANEP_{(NTR,RF)} = ANEP^{TC}_{(NTR,RF)} \text{ x } ANEP^{non-TC}_{(NTR,RF)} \tag{1}$$

$$ANEP_{(NTR,RF)} = \left(1 - AEP^{TC}_{(NTR,RF)}\right) \text{ x } (1 - AEP^{non-TC}_{(NTR,RF)}) \tag{2}$$

The associated return period ($RP$) is calculated as

$$RP_{(NTR,RF)} = \frac{1}{1 - ANEP_{(NTR,RF)}} \tag{3}$$

Therefore, the combined return period (RP) is always closer to the return period of the storm type that contributes the higher exceedance probability (lower return period) at the given (NTR, RF) combination. In this case, that contribution comes from the TC sample.

As this part of the study and the governing equations are discussed in detail in our prior publication (Maduwantha et al., 2024), we don't plan to elaborate on it in the current manuscript to avoid redundancy and additional length. We have added the following text to the main manuscript to clarify this point.

L455: "The joint probability distributions of peak NTR and peak RF of TC and non-TC storms are substantially different. Small to moderate compound events are more frequent in the non-TC storms, whereas the most extreme compound events are more likely generated by TCs. For example, the selected event for demonstrating the event-generation framework (NTR = 1.75 m, 18-h RF = 80 mm) corresponds to a ~106-yr joint return period in the combined joint probability distribution. The same event has a joint return period of ~111 yrs in the TC sample and ~2431 yrs in the non-TC sample, highlighting that rare events are primarily associated with TCs."

[Figure]

**Figure R1**: Kendall's τ between NTR and RF for different RF accumulation times for all events (purple), TCs (orange), and non-TCs (green) for samples conditioned on NTR (a and b) and RF (c and d). Filled markers indicate values that are significant at the 5% level. The black circles with vertical dashed lines show the selected RF accumulation for each location (Adapted from Maduwantha et al., 2024).

[Figure]

**Figure R2**: Results of bivariate statistical analysis for Gloucester City; TC events (left) and non-TC events (right): (a), (b) when conditioning on NTR; (c), (d) when conditioning on RF; and (e), (f) when two conditioning samples are combined. Quantile isolines of the 5, 10, 20, 50, and 100-year joint return periods are shown where the color scale indicates the relative probability of events along the isolines. Note the different x- and y-axis scales for better clarity (Adapted from Maduwantha et al., 2024). The red star indicates the selected synthetic event for the demonstration of events generation.

[Figure]

**Figure R3**: Modified Fig. 3 of the original manuscript. The red star indicates the selected synthetic event for the demonstration of events generation.